# Phylodynamic theory of persistence, extinction and speciation of rapidly adapting pathogens

Le Yan[1]*, Richard A Neher[2]*, Boris I Shraiman[1]*

[1]Kavli Institute for Theoretical Physics, University of California, Santa Barbara, Santa Barbara, United States; [2]Biozentrum, University of Basel, Swiss Institute for Bioinformatics, Basel, Switzerland

**Abstract** Rapidly evolving pathogens like influenza viruses can persist by changing their antigenic properties fast enough to evade the adaptive immunity, yet they rarely split into diverging lineages. By mapping the multi-strain Susceptible-Infected-Recovered model onto the traveling wave model of adapting populations, we demonstrate that persistence of a rapidly evolving, Red-Queen-like state of the pathogen population requires long-ranged cross-immunity and sufficiently large population sizes. This state is unstable and the population goes extinct or 'speciates' into two pathogen strains with antigenic divergence beyond the range of cross-inhibition. However, in a certain range of evolutionary parameters, a single cross-inhibiting population can exist for times long compared to the time to the most recent common ancestor ($T_{MRCA}$) and gives rise to phylogenetic patterns typical of influenza virus. We demonstrate that the rate of speciation is related to fluctuations of $T_{MRCA}$ and construct a 'phase diagram' identifying different phylodynamic regimes as a function of evolutionary parameters.
DOI: https://doi.org/10.7554/eLife.44205.001

## Introduction

In a host population that develops long-lasting immunity, a pathogen can persist by infecting immunological naive individuals such as children, or through rapid antigenic evolution that enables the pathogen to evade immunity and re-infect individuals. Childhood diseases like measles or chicken pox fall into the former category, while influenza virus adapts rapidly and re-infects most humans multiple times during their lifespan. The continuous adaptation of influenza is facilitated by high mutation rates resulting in diverse populations of co-circulating viral strains. Nevertheless, almost always a single variant eventually outcompetes the others such that diversity within one subtype or lineage remains limited (*Petrova and Russell, 2018*).

The contrast of rapid evolution while maintaining limited genetic diversity is most pronounced for the influenza virus subtype A/H3N2. *Figure 1* shows a phylogenetic tree of HA sequences of type A/H3N2 with the characteristic 'spindly' shape. The most recent common ancestor of the population is rarely more than 3–5 years in the past (*Rambaut et al., 2008*). Other pathogenic RNA viruses that typically do not reinfect the same individual, (measles, mumps, HCV, or HIV) diversify for decades or centuries (*Grenfell et al., 2004*). Interestingly, influenza B has split into two co-circulating lineages in the 1970s which by now are antigenically distinct (*Rota et al., 1990*) and maintain intermediate levels of diversity (see *Figure 1*).

Influenza virus infections elicit lasting immunity rendering most individuals non-susceptible to viruses that circulated during their lifetime (*Fonville et al., 2014*). The virus population escapes collective human immunity by accumulating amino acid substitutions in its surface glycoproteins (*Koel et al., 2013*; *Wilson and Cox, 1990*). Extensive genetic characterizations have shown that

*For correspondence:
lyan@kitp.ucsb.edu (LY);
richard.neher@unibas.ch (RAN);
shraiman@kitp.ucsb.edu (BIS)

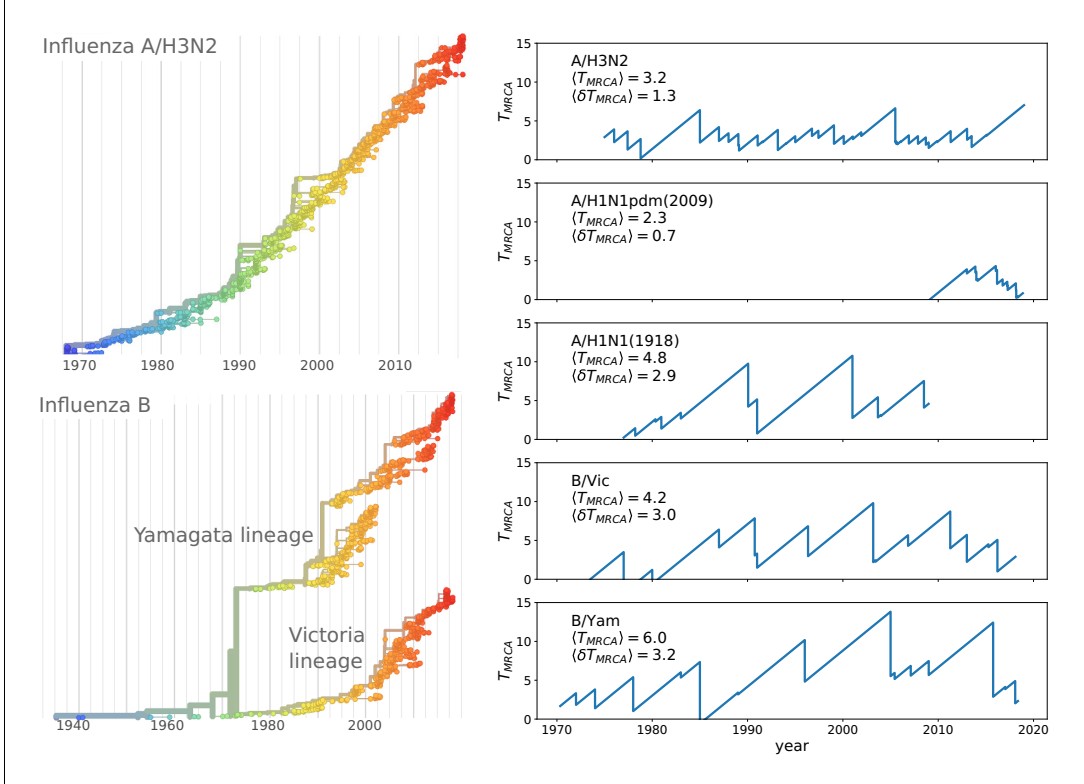

**Figure 1.** Spindly phylogenies and speciation in different human seasonal influenza virus lineages. The top left panel shows a phylogeny of the HA segment of influenza A virus of subtype H3N2 from its emergence in 1968 to 2018. The virus population never accumulates much diversity but is rapidly evolving. The lower left panel shows a phylogeny of the HA segment of influenza B viruses from 1940 to 2018. In the 70s, the population split into two lineages known as Victoria (B/Vic) and Yamagata (B/Yam) in the 1970s. The graphs on the right quantify diversity via the time to the most recent common ancestor $T_{MRCA}$ for different influenza virus lineages. Influenza B viruses harbor more genetic diversity than influenza A viruses. The subtype A/H3N2 in particular coalesces typically in 3y while deeps splits in excess of 5y are rare.

DOI: https://doi.org/10.7554/eLife.44205.002

within each subtype many HA sequence variants co-circulate (*Rambaut et al., 2008*; *Fitch et al., 1997*). These variants differ from each other by ~10 substitutions and compete for susceptible hosts (*Strelkowa and Lässig, 2012*). The rapid sequence evolution results in a decay of immune cross-reactivity over ~10 years (*Smith et al., 2004*; *Bedford et al., 2014*; *Fonville et al., 2014*; *Neher et al., 2016*).

Epidemiological dynamics of influenza is often modeled using generalizations of the classic Susceptible-Infected-Recovered (SIR) model to multiple antigenically distinct viral strains (*Kermack and McKendrick, 1927*; *Gog and Grenfell, 2002*). Such models need to capture (i) how the infection with one strain affects susceptibility to other strains and (ii) how novel strains are generated from existing strains by mutations. A common approach has been to impose a discrete one-dimensional strain space in which new strains are generated by mutation of adjacent strains. Infection results in a reduction of susceptibility in a manner that depends on the distance in this one-dimensional strain space (*Andreasen et al., 1996*; *Gog and Grenfell, 2002*). Such models naturally result in 'traveling waves' in the sense that the pathogen population moves through strain space by recurrent emergence of antigenically advanced variants produced by mutation from neighboring strains (*Lin et al., 2003*).

These models of antigenically evolving populations are related to general models of rapid adaptation in which populations form a traveling wave moving towards higher fitness (*Tsimring et al., 1996*; *Rouzine et al., 2003*; *Desai and Fisher, 2007*; *Neher et al., 2014*), reviewed in *Neher (2013a)*. Recently, *Rouzine and Rozhnova (2018)* described an explicit mapping between a SIR model in a one-dimensional antigenic space and traveling wave models in fitness.

Traveling wave (TW) models in a one-dimensional antigenic space naturally result in spindly phylogenies: There is only one possible direction for immune escape and the fastest growing most antigenically advanced strain grows drives *all* other strains extinct. Influenza viruses, however, can escape immunity by mutations at a large number of positions (*Wilson and Cox, 1990*), suggesting antigenic space is high dimensional (*Perelson and Oster, 1979*). In many dimensions, different viral strains can escape immunity via different paths and diverge sufficiently from each other until they no longer compete for hosts and thereafter propagate independently evolve. A satisfactory explanation of spindly phylogenies therefore has to describe how evolution in a high dimensional space reduces to an effectively one-dimensional path without persistent branching or rapid extinction. Several computational studies have addressed this question and identified cross-immunity (*Bedford et al., 2012*; *Tria et al., 2005*; *Koelle et al., 2011*; *Ferguson et al., 2003*; *Sasaki and Haraguchi, 2000*) as well as deleterious mutations (*Koelle and Rasmussen, 2015*; *Gog and Grenfell, 2002*) as critical parameters. We will discuss this earlier work at greater length below.

Our work aims to examine the conditions under which the evolving pathogen can maintain a spindly phylogeny with an approximately constant level of diversity – sufficient to avoid extinction, yet constrained from further branching by cross-inhibition between not too distant strains. We show that long range cross immunity in generic stochastic models of antigenic evolution generates such phylogenies. However, in the long term the viral population either 'speciates' into weakly interacting diverging lineages or goes extinct with rates that are controlled by three dimensionless combinations of model parameters. While the relation of these parameters to the known characteristics of influenza epidemiology and evolution is not direct, the general 'phase diagram' captured by the parameters of the simple model illustrates the key competing factors governing expected long-term dynamics.

## Results

### Model

A model of an antigenically evolving pathogen population needs to account for cross-immunity between strains and the evolution of antigenically novel strains. We use an extension of the standard multi-strain SIR model (*Gog and Grenfell, 2002*). The fraction of individuals $I_a$ infected with viral strain $a$ changes according to

$$\frac{d}{dt}I_a = \beta S_a I_a - (\nu + \gamma)I_a \tag{1}$$

where $\beta$ is the transmissibility, $S_a$ is the population averaged susceptible to strain $a$, $\nu$ is the recovery rate, and $\gamma$ is the population turnover rate. The fraction $R_a$ of the population recovered from infection with strain $a$ changes according to

$$\frac{d}{dt}R_a = \nu I_a - \gamma R_a \tag{2}$$

Our focus here is on antigenically evolving pathogens that reinfect an individual multiple times during its life-time, we shall ignore population turnover and set $\gamma = 0$ right away to simplify presentation.

The dynamics of $I_a$ depends on the average susceptibility of the host population $S_a = \langle S_a(i) \rangle_i$, while the susceptibility $S_a(i)$ of host $i$ depends on the host's history of previous infections. A plausible representation of the history dependence of susceptibility at the level of individuals has a product form (*Wikramaratna et al., 2015*)

$$S_a^{(\sigma)} = \langle \prod_b (1 - K_{ab}\sigma_b(i)) \rangle_i \tag{3}$$

where $\sigma_b(i)$ is one or zero depending on whether host $i$ has or has not been previously infected with strain $b$. Matrix $K_{ab} \leq 1$ quantifies the cross-immunity to strain $a$ due to prior infection with strain $b$. Thus, *Equation 3* expresses the susceptibility $S_a$ in terms of a product of attenuation factors each arising from a prior infection by a different strain $b$. A simple, but adequate approximation for the

population averaged susceptibility is provided by replacing $\sigma_b(i)$ in the product in **Equation 3** by the fraction of the population $R_b$ that recovered from infection with strain $b$:

$$S_a \approx \prod_b (1 - K_{ab}R_b) \approx e^{-\sum_b K_{ab}R_b} \tag{4}$$

This corresponds to the 'order one independence closure' by **Kryazhimskiy et al. (2007)** and is known as Mean-Field approximation in physics (**Weiss, 1907**; **Landau and Lifshitz, 2013**). The Mean-Field approximation here corresponds to ignoring correlations between subsequent infection in the individual histories. Approximating the product by the exponential is justified because the total fraction of the host population infected by any single strain in the endemic regime is typically small (**Yang et al., 2015**). A detailed derivation of **Equation 4** and more detailed discussion of approximations is given in Appendix 1. While the original formulation of immunity in **Equation 3** is based on the infection history of individuals (**Andreasen et al., 1997**), the population average over the factorized distribution of histories relates the model to status based formulations (**Gog and Grenfell, 2002**). While some differences between status and history-based models have been reported (**Ballesteros et al., 2009**), others have shown that different model types have similar properties (**Ferguson and Andreasen, 2002**). The differences between these models and approximations are small compared to the crudeness with which these simple mathematical models capture the complex immunity profile of the human population. A model similar to ours has been successfully applied to influenza virus evolution (**Luksza and Lässig, 2014**).

We note that differentiating **Equation 4** with respect to time defines the equation governing the dynamics of population average susceptibility

$$\frac{d}{dt}S_a = -\nu S_a \sum_b K_{ab}I_b \tag{5}$$

which is exactly the same as the dynamics of susceptibility in **Gog and Grenfell (2002)** and **Luksza and Lässig (2014)** in the limit of negligible population turnover $\gamma/\nu \ll 1$.

New strains are constantly produced by mutation with rate $m$. The novel strain will differ from its parent at one position in its genome. Following **Luksza and Lässig (2014)**, we assume that cross-immunity decays exponentially with the number of mutations that separate two strains:

$$K_{ab} = \alpha e^{-\frac{|a-b|}{d}} \tag{6}$$

where $|a-b|$ denotes the mutational distance between the two strains, $d$ denotes the radius of cross-immunity measured in units of mutations. Antigenic space is thereby assumed to be high dimensional and antigenic distance is proportional to genetic distance in the phylogenetic tree (**Neher et al., 2016**). The parameter $\alpha \leq 1$ quantifies the reduction of susceptibility to reinfection by the same strain and hence the overall strength of protective immunity. We shall set $\alpha = 1$ corresponding to perfect protection here for simplicity of presentation. Our analysis below applies equally well to the more realistic case of $\alpha < 1$, since in our approximation this parameter can be eliminated by rescaling $R_a$ and $I_a$ and ultimately merely renormalizes the host population size, which serves as one of the 'control parameters' in our analysis.

Cross-immunity and the mutation/diversification process are illustrated in **Figure 2**. An infection with a particular strain (center of the graph)

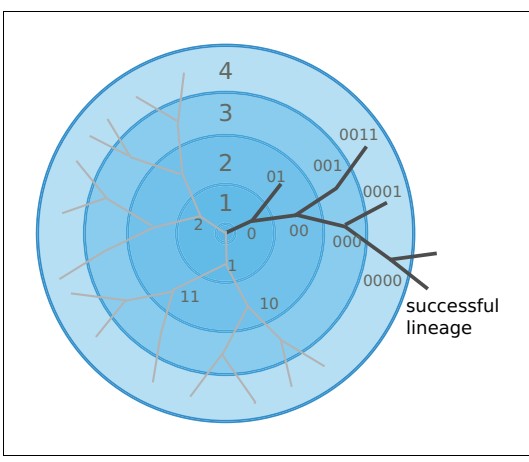

**Figure 2.** Viral populations escape adaptive immunity by accumulating antigenic mutations. Via cross-reactivity, the immunity foot-print of ancestral variants (center of the graph) mediates competition between related emerging viral strains and can drive all but one of the competing lineages extinct. At high mutation rates and relatively short range of antigenic cross-reactivity, different viral lineages can escape inhibition and continue to evolve independently.
DOI: https://doi.org/10.7554/eLife.44205.003

generates a cross-immunity footprint (shaded circles). Mutation away from the focal strain reduces the effect of existing immunity in the host population, but complete escape requires many mutations. Hence closely related viruses compete against each other for susceptible individuals.

The above model was formulated in terms of the deterministic *Equations 1-4*. The actual dynamics, however, is stochastic in two respects: (i) antigenic mutations are generated at random with rate $m$ and (ii) stochasticity of infection and transmission. The latter can be captured by interpreting the terms in *Equation 1* as rates of discrete transitions in a total population of $N_h$ hosts. This stochasticity is particularly important for novel mutant strains that are rare. Most rare strains are quickly lost by chance even if they have a growth advantage due to antigenic novelty. To account for stochasticity in a computationally efficient way, we employ a clone-based hybrid scheme where mutation and the dynamics of rare mutants are modeled stochastically, while common strains follow deterministic dynamics, see Materials and methods (Clone-based simulations).

We will use the recovery rate $\nu$ to set the unit of time, fixing $\nu = 1$ in rescaled units. The remaining parameters of the model are (1) the transmission rate $\beta$ - in our units the number of transmission events per infection and hence equal to the basic reproduction number $R_0$, (2) the mutation rate $m$, (3) the range of cross-immunity $d$ measured as the typical number of mutations needed for an $e$-fold drop of cross-inhibition, and (4) the host population size $N_h$.

## Phenomenology

Before proceeding with a quantitative analysis we discuss different behaviors qualitatively. *Figure 3A* shows several trajectories of prevalence $I_{tot} = \sum_a I_a$ (i.e. total actively infected fraction) for several different parameters. Depending on the range of cross-immunity, the pathogen either goes extinct after a single pandemic (red line) or settles into a persistently evolving state, the Red Queen State (RQS) traveling wave (*Van Valen, 1973* In large populations the RQS exhibits oscillations in prevalence. As we will discuss further below, the RQS state is transient and either goes extinct after some time or splits into multiple antigenically diverging lineages that propagate independently. To quantitatively understand the dependence on parameters, we will further simplify the model and establish a connection to models of rapid adaptation in population genetics. *Figure 3BC* shows parameter regimes corresponding to distinct qualitative behaviors. The relevant parameters are three combinations of the population size $N_h$, the selection coefficient of novel mutations $s$, the mutation rate $m$, and the radius of cross-immunity $d$. A long-lived but transient RQS regime is flanked be the regime of deterministic extinction (red) and the regime of continuous branching and diversification – the 'speciation' regime (blue). The RQS regime itself undergoes a transition from a steady traveling wave (yellow) to a limit cycle oscillation (green) with increasing population size. The location of the boundaries depend on the time scale of observation as the cumulative probability of extinction and speciation increases with time.

## Large effect antigenic mutations allow transition from pandemic to seasonal dynamics

A novel virus in a completely susceptible population will initially spread with rate $\beta - 1$ and the pandemic peaks when susceptible fraction falls to $\beta^{-1}$. The trajectory of such a pandemic strain in the time-susceptibility plane is indicated in red in *Figure 3D*. Further infections in the contracting epidemic will then push susceptibility below $\beta^{-1}$ – the propagation threshold for the virus – and without rapid antigenic evolution the pathogen will go extinct after a time $t \sim \beta^{-1} \log N_h$. Such boom-bust epidemics are reminiscent of the recent Zika virus outbreak in French Polynesia and the Americas where in a short time a large fraction of the population was infected and developed protective immunity (*O'Reilly et al., 2018*).

Persistence and transition to an endemic state is only possible if the pathogen can evade the rapid build-up of immunity via a small number of large effect antigenic mutations. This process is indicated in *Figure 3D* by horizontal arrows leading to antigenically evolved strains of higher susceptibility and bears similarity to the concept of 'evolutionary rescue' in population genetics (*Gomulkiewicz, 1995*). The parameter range of the idealized SIR model that avoid extinction after a pandemic resulting in persistent endemic disease is relatively small. Yet, various factors like geographic structure, heterogeneity of host adaptation and population turn-over slow down the pandemic and extinction, thereby increasing the chances of sufficient antigenic evolution to enter the endemic,

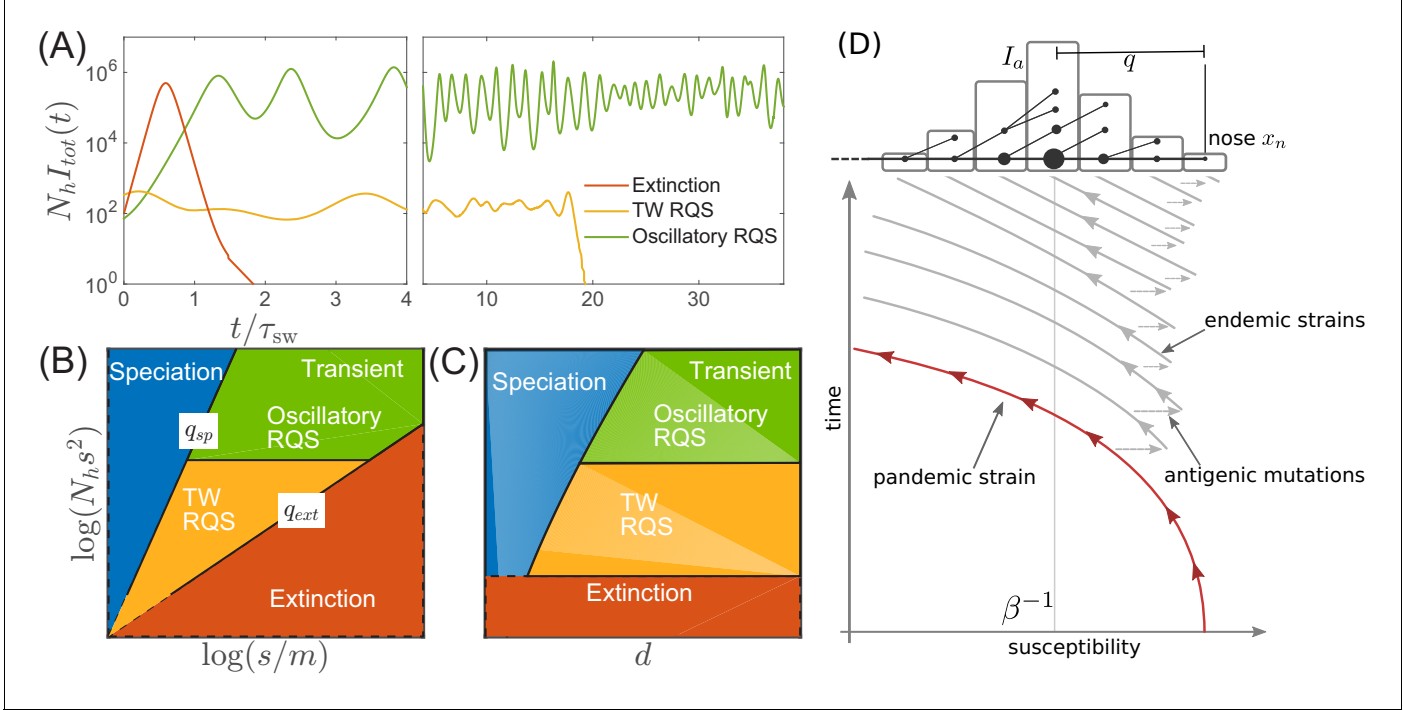

**Figure 3.** Extinction, speciation, and oscillations in multi-strain SIR models. (A) Typical trajectories of infection prevalence in the parameter regimes corresponding to extinction (red), traveling wave RQS (yellow) and oscillatory RQS (green). Panels B and C schematically show parameter regimes corresponding to these qualitatively different behaviors. As explained in the text the boundaries $q_{ex}$, $q_{sp}$ of the RQS regime depend explicitly on the time scale considered (see also **Figure 7**). Simulation results supporting this diagram are shown in **Figure 3—figure supplements 1** and **2**. Panel (D) is a schematic illustrating how a novel pandemic strain (red) can settle into an endemic RQS state. As the cumulative number of infected individuals increases, the susceptible fraction decreases, and survival of the strain depends on the emergence of antigenic escape mutations (gray). The top part of the panel illustrates the population composition at a particular time point. Rare pioneering variants are $q$ mutations ahead of the dominant variant and grow with rate $x_n$. (Note, that boundaries of the 'extinction' regime in (B,C) correspond to $q$ value close to one.) Different lineages are related via their phylogenetic tree embedded in the fitness distribution in the population.

DOI: https://doi.org/10.7554/eLife.44205.004

The following figure supplements are available for figure 3:

**Figure supplement 1.** 'Phase diagram' in the clone-based simulation.

DOI: https://doi.org/10.7554/eLife.44205.005

**Figure supplement 2.** Value of $q$ across the 'Phase diagram'.

DOI: https://doi.org/10.7554/eLife.44205.006

RQS-type, regime. The 2009 pandemic influenza A/H1N1 has undergone such a transition from a pandemic to a seasonal/endemic state. We shall not investigate the transition process in detail here, but will assume that endemic regime has been reached.

## Long-range cross-immunity results in evolving but low diversity pathogen populations

Once the pathogen population has established an endemic circulation through continuous antigenic evolution (green and yellow regimes in **Figure 3BC**), the average rate of new infections $\beta \sum_a I_a S_a / I_{tot}$ fluctuates around the rate of recovery $\nu = 1$ (in our time units). This balance is maintained by the steady decrease in susceptibility due to rising immunity against resident strains and the emergence of antigenically novel strains, see **Figure 3D**. If the typical mutational distance between strains is small compared to the cross-immunity range $d$, the rate at which susceptibility decreases is similar for all strains. To see this we expand $K_{ab}$ in **Equation 5**

$$S_a^{-1} \frac{d}{dt} S_a(t) = -\sum_b e^{-\frac{|a-b|}{d}} I_b \approx -I_{tot} + \sum_b \frac{|a-b|}{d} I_b \qquad (7)$$

where we have used that $|a-b| \ll d$ for all pairs of strains with substantial prevalence. In fact it will suffice to keep only the first, leading, term on the right hand side. Close to a steady state, prevalent strains obey $\beta S_a \approx 1$. We can hence define the instantaneous growth rate of strain $x_a = (\beta S_a - 1) \ll 1$ as its effective fitness. In this limit, the model can be simplified to

$$\frac{d}{dt}I_a = x_a I_a$$
$$\frac{d}{dt}x_a \approx -I_{tot}$$

(8)

The second equation means that effective fitness of all strains $a$ decreases approximately at the same rate since the pathogen population is dominated by antigenically similar strains.

If a new strain $c$ emerged from strain $a$ by a single antigenic mutation, its mutational distance from a strain $b$ is $|c-b| = |a-b|+1$ and $K_{cb} = K_{ab}e^{-d^{-1}} \approx K_{ab}(1-d^{-1})$. The population susceptibility of strain $c$ is therefore increased to

$$S_c \approx e^{-(1-d^{-1})\sum_b K_{ab}R_b} \approx S_a\left(1 - \frac{\log S_a}{d}\right)$$

(9)

Since the typical susceptibility is of order $\beta^{-1}$, the growth rate of the mutant strain $c$ is $s = d^{-1}\log\beta$ higher than that of its parent. The growth rate increment, $s$, plays the role of a selection coefficient in typical population genetic models and corresponds to the step size of the fitness distribution in *Figure 3D*. In such models, individuals within a fitness class (bin of the histogram) are equivalent and different classes ≪ be modeled as homogeneous populations which greatly accelerates numerical analysis of the model, see Materials and methods.

*Rouzine and Rozhnova (2018)* have recently formulated a similar model of antigenic evolution of rapidly adapting pathogens. Analogously to our model, Rouzine and Rozhnova couple strain dynamics to antigenic adaptation through mutations, albeit assuming a one-dimensional antigenic space. In agreement with Rouzine and Rozhnova, we find that selection coefficients of novel mutations are inversely proportional to the cross-immunity rate $d$ and increase with infectivity $\beta$, see *Equation 9*. Rouzine and Rozhnova, however, do not consider oscillations, extinction, and speciation (see below).

The simplified model in *Equation 8*, along with the model developed by *Rouzine and Rozhnova (2018)*, is analogous to the traveling wave (TW) models of rapidly adapting asexual populations that have been studied extensively over the past two decades (*Tsimring et al., 1996*; *Desai and Fisher, 2007*; *Rouzine et al., 2003*; *Hallatschek, 2011*), see *Neher (2013a)* for a review. These models describe large populations that generate beneficial mutations rapidly enough that many strains co-circulate and compete against each other. The fittest (most antigenically advanced) strains are often multiple mutational steps, $q$, ahead of the most common strains, see *Figure 3D*. This 'nose' of the fitness distributions contains the strains that dominate in the future and the only adaptive mutations that fixate in the population arise in pioneer strains in the nose. Consequently, the rate with which antigenic mutations establish in the population is controlled by the rate at which they arise in the nose (*Desai and Fisher, 2007*). If the growth rate at the nose of the distribution, $x_n$, is much higher than antigenic mutation rate, $x_n \gg m$, it takes typically

$$\tau_a = \frac{\log(x_n/m)}{x_n}$$

(10)

generations before a novel antigenic mutation arises in a newly arisen pioneer strain that grows exponentially with rate $x_n$. The advancement of the nose is balanced rapidly by the increasing population mean fitness.

If beneficial mutations have comparable effects on fitness and population sizes are sufficiently large ($Nm \gg 1$), the fitness distribution has an approximately Gaussian shape with a variance $\sigma^2 \approx 2s^2\log(Ns)/\log^2(x_n/m)$. The wave is $\sigma/s$ mutations wide, while the most advanced strains are approximately $q = 2\log(Ns)/\log(x_n/m)$ ahead of the mean (*Desai and Fisher, 2007*). Two contemporaneous lineages coalesce on a time scale $\tau_{sw} = sq/\sigma^2 = s^{-1}\log(x_n/m)$ and the branching patterns of the tree resemble a Bolthausen-Sznitman coalescent rather than a Kingman coalescent (*Desai et al., 2013*; *Neher and Hallatschek, 2013b*).

In circulating influenza viruses, typically around 3–10 adaptive mutations separate pioneer strains from the most common variants (*Strelkowa and Lässig, 2012*; *Neher and Bedford, 2015*). While this clearly corresponds to a regime where multiple stains compete, it does not necessarily mean that asymptotic formulae assuming $q \gg 1$ are accurate. Nevertheless, many qualitative features of TW models have been shown to qualitatively extend into regimes where $q$ takes intermediate values (*Neher and Hallatschek, 2013b*).

While parameter $N$ in the TW models summarized above is a fixed population size, the corresponding entity in our SIR model is the fluctuating pathogen population size $N_p$ which is related to the (fixed) host population size $N_h$ by $N_p = N_h I_{tot}$. The average $I_{tot}$ depends on other parameters of the model, scaling in particular with $\bar{I} \sim s^2$. Hence, it will be convenient for us to use $N_h s^2$ as one of the relevant 'control parameters', replacing $N$ of the standard TW model.

## Stability and fluctuations of the RQS

In contrast to most population genetic models of rapid adaptation, our epidemiological model does not control the total population size directly. Instead, the pathogen population size (or prevalence) depends on the host susceptibility, which in itself is determined by recent antigenic evolution of the pathogen. The coupling of these two different effects results in a rich and complicated dynamics (see *Figure 4A* for an example trajectory): The first effect is ecological: a bloom of the pathogen depletes susceptible hosts leading to a crash in pathogen population and a tendency of the population size to oscillate *London and Yorke, 1973* (blue line in *Figure 4A*). The second effect is evolutionary: higher nose fitness $x_n$ begets faster antigenic evolution and vice versa, resulting in an apparent instability in the advancement of the antigenic pioneer strains (*Fisher, 2013*) (yellow line and inset in *Figure 4A*). In our epidemiological model, as we shall show below, fluctuations in the rate of antigenic advance of the pioneer strains couple with a delay of $\tau_{sw}$ to the ecological oscillation.

To recognize the ecological aspect of the oscillatory tendency, consider the total prevalence $I_{tot}$ and the mean fitness of the pathogen $X = \sum_a x_a I_a / I_{tot}$

$$\frac{d}{dt} I_{tot} = X I_{tot}; \quad \frac{d}{dt} X = \sigma^2 - I_{tot} \tag{11}$$

which follows directly from *Equation (8)*. Selection on fitness variance $\sigma^2$ increases $X$, while prevalence $I_{tot}$ reduces susceptibility and hence $X$. At fixed variance $\sigma = \bar{\sigma}$ this system is equivalent to a non-linear oscillator describing a family of limit cycles oscillating about $I_{tot} = \bar{\sigma}^2$ and $X = 0$ as shown in *Figure 4B*.

While *Equation (11)* describes the behavior of common strains, the mutation driven dynamics of the antigenic pioneer strains is governed by the equation for $x_n$ that in a continuum limit (suitable for the limit of high mutation rate) reads:

$$\frac{d}{dt} x_n = \tau_{sw}^{-1} x_n - I_{tot} + s\xi(t) \tag{12}$$

The first term on the right hand side represents the rate at which antigenic pioneer strains enter the population, $\tau_a^{-1}$, advancing the nose fitness by an increment $s$ (with $\tau_a^{-1} s = \tau_{sw}^{-1} x_n$ ). The second term on the right hand side of *Equation (12)* represents gradual reduction of susceptibility of the host population, and $\xi(t)$ is a random noise variable representing the stochasticity of the establishment of new strains. The Gaussian white noise $\xi(t)$ is defined statistically by its correlation function $\langle \xi(t)\xi(0) \rangle = \tau_a^{-1} \delta(t)$, see Materials and methods (Stochastic differential-delay simulation).

The first term of *Equation (12)* captures the apparent instability of the nose: an advance of the nose to higher $x_n$ accelerates its rate of advancement. The stabilizing factor is the subsequent increase in $I_{tot}$, but to see how that comes about we must connect *Equation (12)* to *Equation (11)*. The connection is provided by $\sigma^2$ since it is controlled by the emergence of novel strains, that is the dynamics of the 'nose' $x_n$, which impacts the bulk of the distribution after a delay $\tau_{sw}$. Based on the analysis detailed in the Appendix 2, we approximate

$$\sigma^2(t) \approx \tau_{sw}^{-1} x_n(t - \tau_{sw}) \tag{13}$$

relating population dynamics, *Equation (11)*, to antigenic evolution of pioneer strains described by

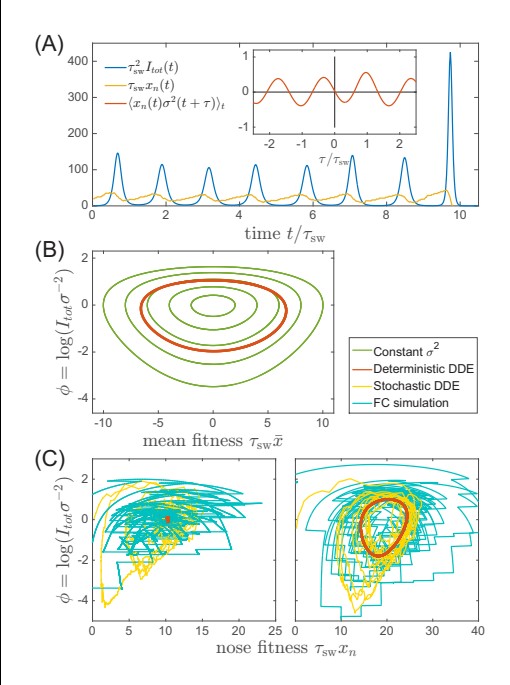

**Figure 4.** Oscillations in antigenically evolving populations. (**A**) An example of the stochastic limit cycle trajectory from the fitness-class simulation. Note the rapid rise and fall of infection prevalence (blue), which causes a drop in nose fitness (yellow) which subsequently recovers (approximately linearly) during the remainder of the cycle. Fluctuations in $I_{tot}(t)$ and $x_n(t)$ from cycle to cycle are caused by the stochasticity of $x_n$, that is antigenic evolution in pioneer strains. A particularly large fluctuation about $\tau_{sw}$ prior to the end, caused a large spike in prevalence, followed by the collapse of $x_n$ below zero and complete extinction. Inset (red) shows the cross-correlation between $x_n$ and $\sigma^2$ which peaks with the delay $\tau = \tau_{sw}$ (additional peaks reflect the oscillatory nature of the state and are displaced by integer multiples of mean period). (**B**) A family of limit cycles in the infection prevalence/mean fitness plane as described by **Equation (11)** with fixed variance. The variation of $\sigma$ governed by the **Equations 11 and 12** (in the deterministic limit) reduces the family to a single limit cycle (red); (**C**) Trajectories in the infection prevalence/nose fitness generated by the stochastic DD system in the regime above (right panel) and below (left panel) the oscillatory instability of the deterministic DD system.
DOI: https://doi.org/10.7554/eLife.44205.007

The following figure supplements are available for figure 4:

**Figure supplement 1.** Amplitude and time scale of oscillations.
DOI: https://doi.org/10.7554/eLife.44205.008
**Figure supplement 2.** Speed of the traveling wave.
DOI: https://doi.org/10.7554/eLife.44205.009

**Equation (12)**. Taken together **Equations (11-13)** define a Differential Delay (DD) system of equations. Sample simulations of this stochastic DD system are shown in **Figure 4 BC**. The delay approximation **Equation (13)** is supported by the cross-correlation of $x_n(t)$ and $\sigma^2(t')$ measured using fitness-class simulations (see **Figure 4A** Inset).

The deterministic limit of the DD system (obtained by omitting the noise term in **Equation (12)**) has a fixed point at $\tau_{sw}^{-1}\bar{x}_n = \bar{\sigma}^2 = 2\tau_{sw}^{-2}\log(N_h\bar{I})$. Small deviations decay in underdamped oscillations with frequency $\omega = \bar{\sigma} = \tau_{sw}^{-1}\sqrt{2\log(N_h\bar{I})}$ if $\omega\tau_{sw} < 2\pi$. For $\omega\tau_{sw} > 2\pi$, the system fails to recover from a deviation of the nose in a single period and the steady state becomes unstable to a limit cycle oscillation. The nonlinearity of **Equation (11)** implies a longer period with increasing amplitude and the system is stabilized at a limit cycle with the period long enough compared to the feedback delay $\tau_{sw}$. In Appendix 3, we derive the threshold of oscillatory instability to lie at $\log(N_h\bar{I}_{osc}s) \approx 8.3$ (leading to limit cycle period $T \approx 1.5\tau_{sw}$, see **Figure 4—figure supplement 1**). We also find that the amplitude of the oscillation $\log(I_{max}/\bar{I})$ scales as $\log(N_h\bar{I})$ for large values of the later. This transition defines quantitatively the boundary between the TW RQS and the Oscillatory RQS regimes that appear on the phase diagrams in **Figure 3 (BC)**. The validity of the predictions of standard TW theory for our adapting SIR system are explored in **Figure 4—figure supplement 2**.

The distinction between the TW and Oscillatory RQS is obscured by the stochasticity of antigenic advance, **Equation (12)**, which continuously feeds the underdamped relaxation mode, generating a noisy oscillation with the frequency $\omega$ defined above. The difference between the two regimes is illustrated by **Figure 4C**: in the TW RQS noisy oscillation is about the fixed point, whereas in the Oscillatory RQS it is about deterministic limit cycle.

Interestingly, the dynamics of the Oscillatory RQS, as shown in **Figure 4A**, can be understood in terms of a non-linear relaxation oscillator. At relatively low infection prevalence nose fitness $x_n$ increases until rising $I_{tot}$ catches up with it (when $I_{tot} = \tau_{sw}^{-1}x_n$) driving it down rapidly. Once this 'mini-pandemic' burns out, the population returns to the low prevalence part of the cycle $I_{tot} < \tau_{sw}^{-1}x_n$, when $x_n$ begins to increase again.

## The rate of extinction

While in the deterministic limit the differential-delay system predicts a stable steady TW for $q > q_{ex}$, $\bar{I} < \bar{I}_{osc}$ and a limit cycle above $\bar{I}_{osc}$, fluctuations in the establishment of the antigenic pioneer strains (*Equation (12)*) can lead to stochastic extinction. In fact, both the TW and Oscillatory RQS (see *Figure 3BC*) are transient, subject to extinction due to a sufficiently large stochastic fluctuation. (Note however the contrast with the 'extinction' state in *Figure 3BC*, where extinction is deterministic and rapid.) The rate of extinction depends on $q$ and $\log(N_h\bar{I})$ as shown in *Figure 5A*. The time to extinction increases dramatically in the range of $q \sim 1 - 2$ and more slowly thereafter. Although extinction is fluctuation driven, the mechanism of extinction in the oscillatory state is related closely to the deterministic dynamics, according to which large amplitude excursion in infection prevalence can lead to extinction. A large $x_n$ advance leads, after a time $\tau_{sw}$ to a rise in prevalence $I_{tot}$, followed by the rapid fall in the number of susceptible hosts and hence loss of viral fitness. This turns out to be the main mode of fluctuation driven extinction as illustrated by *Figure 4C*. One expects extinction to take place when a fluctuation induced deviation $\delta x$ of the fitness of pioneer strains becomes of the order of the mean $\bar{x}_n$. New mutations at the nose accumulate with rate $1/\tau_a$ such that at short times $t$ we expect $\delta x \approx s\sqrt{t/\tau_a}$. Hence $\delta x$ becomes of the order of the mean $\bar{x}_n$ at times $\tau_{ext} \sim q\tau_{sw}$. However the probability of extinction will also depend on the shape of the oscillatory limit cycle (as it depends on the minimum of infection prevalence during the cycle), which in turn depends on $\log(N_h\bar{I})$. Numerical simulations, *Figure 5B*, confirm the dependence of $\tau_{ext}$ on $q$ and $\log(N_h\bar{I})$. We note that the rate increase in $\tau_{ext}$ with increasing $q$ slows down in the oscillatory regime and appears

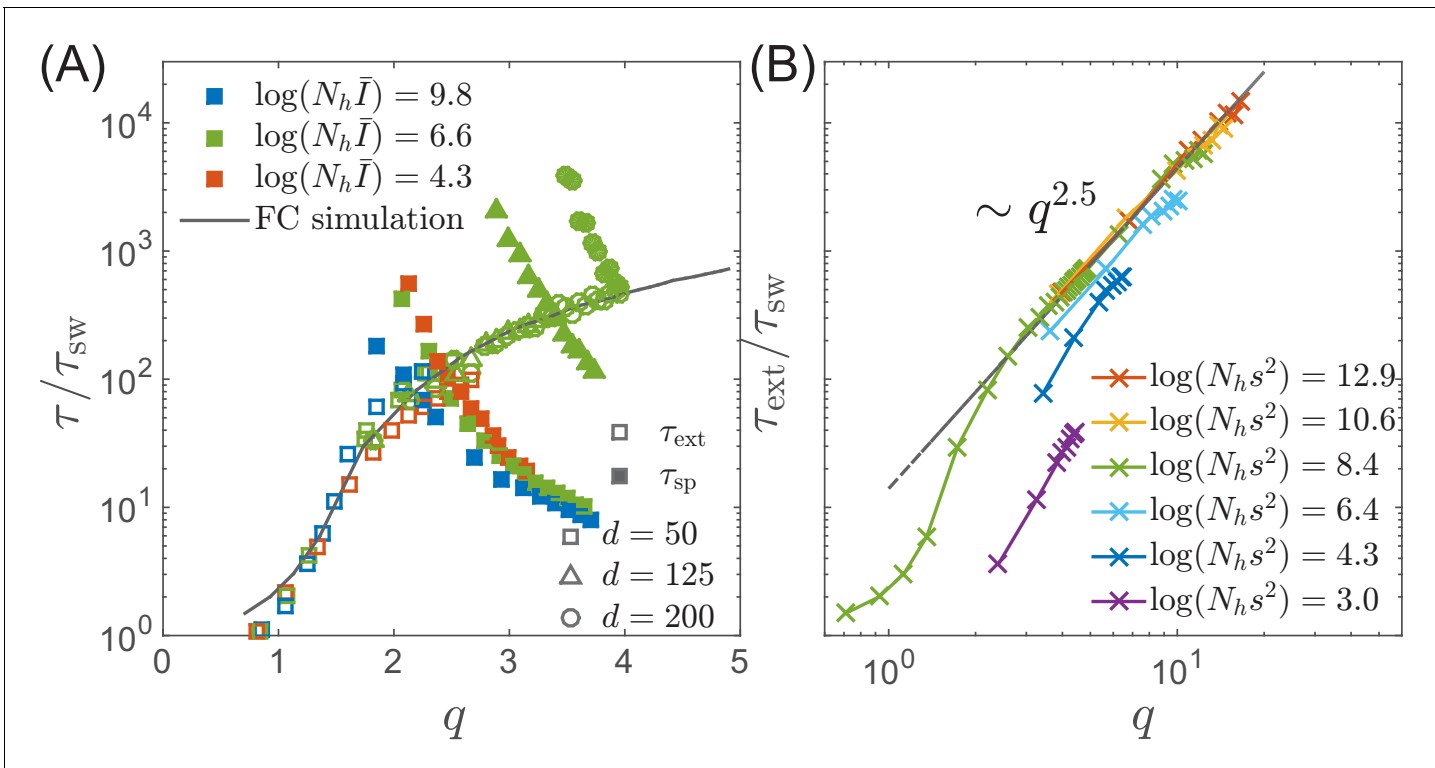

**Figure 5.** Extinction and speciation dynamics. (A) Simulation results for the average extinction time $\tau_{ext}$ (open symbols) and the average speciation time $\tau_{sp}$ (filled symbols) as a function of pathogen diversity $q$. The life time of the endemic RQS state is limited by the smaller of $\tau_{ext}$ and $\tau_{sp}$. If $\tau_{sp} < \tau_{ext}$, the population tends to speciate and persist, while the population is more likely to go extinct if $\tau_{sp} > \tau_{ext}$. The graph shows $\tau_{sp}$ and $\tau_{ext}$ rescaled with the sweep time $\tau_{sw}$ as a function of genetic diversity measured by the number of mutations $q$. The speciation time $\tau_{sp}$ increases with the range of cross-inhibition ($d$) and decreases with $q$. Note the agreement between the results of the fitness class-based simulation (black line in (A)) and the clone-based simulation (colored squares in (A)). (B) Extinction time over a broad range of parameters, obtained via fitness class-based simulation of population dynamics, confirms its primary dependence on $q$ for large population sizes.
DOI: https://doi.org/10.7554/eLife.44205.010

to approach a power law dependence $\tau_{\text{ext}}/\tau_{\text{sw}} \sim q^{2.5}$ (albeit over a limited accessible range): presently we do not have an analytic understanding of this specific functional form.

## The rate of speciation

The correspondence of the multi-strain SIR and the TW models discussed above assumes that cross-immunity decays slowly compared to the coalescent time of the population, that is $d/q \gg 1$. In this case, all members of the population compete against each other for the same susceptible hosts. Conversely, if the viral population were to split into two sub-populations separated by antigenic distance greater than the range of cross-inhibition $d$, these sub-population would no-longer compete for the hosts, becoming effectively distinct viral 'species' that propagate (or fail) independently of each other. Such a split has for example occurred among influenza B viruses, see *Figure 1*.

A 'speciation' event corresponds to a deep split in the viral phylogeny, with the $T_{MRCA}$ growing without bounds, see *Figure 1* and *Figure 6A*. This situation contrasts the phylogeny of the single competing population, where $T_{MRCA}$ fluctuates with a characteristic ramp-like structure generated by stochastic extinction of one of the two oldest clades. In each such extinction event the MRCA jumps forward by $\delta T$. Hence the probability of speciation depends on the probability of the two oldest clades to persist without extinction for a time long enough to accumulate antigenic divergence in excess of $d$. The combined carrying capacity of the resulting independent lineages is then twice their original carrying capacity as observed in simulations, see *Figure 6B*.

To gain better intuition into this process let's follow two most antigenically advanced 'pioneer strains'. In the TW approximation one of these will with high probability belong to the backbone giving the rise to the persisting clade, while the other clade will become extinct, unless it persist long enough to diverge antigenically beyond $d$, becoming a speciation event. As their antigenic distance gradually increases, the two clades are evolving to evade immunity built up against the common ancestor. The less advanced of the two clades is growing less rapidly and takes longer to generate antigenic advance mutations, resulting in still slower growth and slower antigenic advance. Deep splits are hence unstable and it is rare for a split to persist long enough for speciation. In Appendix 5, we reformulate this intuition mathematically as a 'first passage'-type problem which shows that $T_{MRCA}$ distribution has an exponential tail which governs the probability of speciation events.

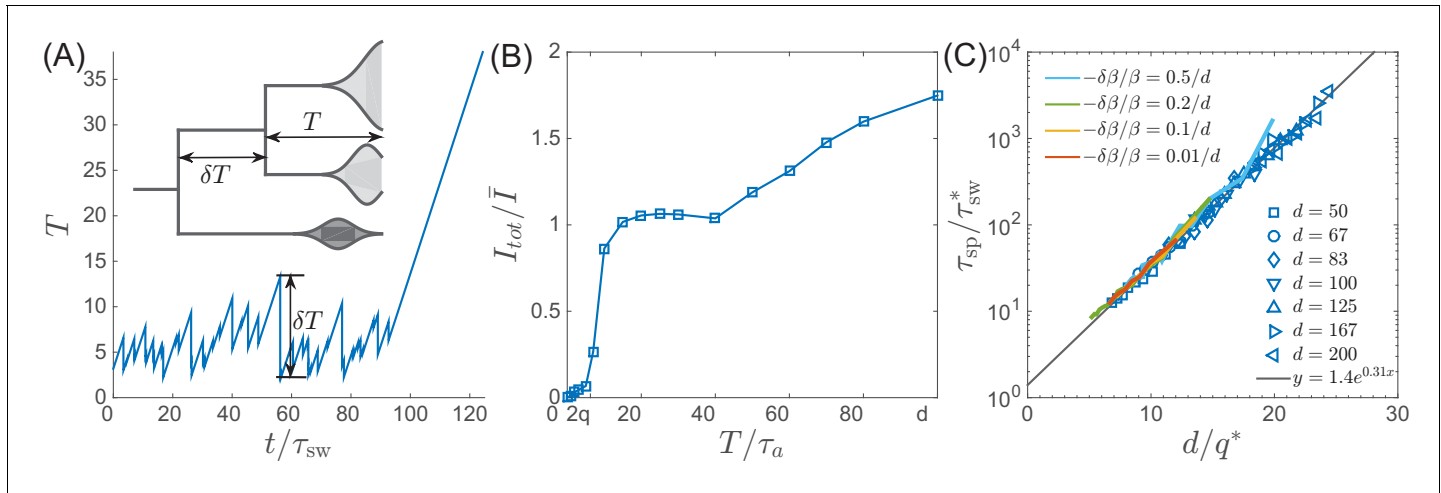

**Figure 6.** Speciation into antigenically distinct lineages. (**A**) To speciate, two lineage have to diverge enough to substantially reduce cross-reactivity, that is $T$ needs to be comparable to $d$. Inset: Illustration of the definition of time to most recent common ancestor $T$ and the time interval $\delta T$ by which $T$ advances. (**B**) If such speciation happens, the host capacity - the average number of infected individuals increases two-fold. (**C**) The probability of such deep divergences decreases exponentially with the ratio $d/q^*$, where effective antigenic diversity is $q^* = 2\log(N_h s^2)/\log(s/m)$. In the presence of deleterious mutations, the relevant $q$ is not necessarily the total advance of the pioneer strains, but only the antigenic contribution. This antigenic advance $q^*$ can be computed as $q^* = \sqrt{2\log(N_h s^2)\sigma_{ag}^2}$ with antigenic variance $\sigma_{ag}^2 = \sigma^2 - \sigma_\beta^2$, where $\sigma_\beta^2$ is fitness variance due to deleterious mutations. With this correction, speciation times agree with the predicted dependence (colored lines).

DOI: https://doi.org/10.7554/eLife.44205.011

*Figure 6C* shows that the time to speciation increases approximately exponentially with the ratio $d/q$. More precisely we found that average simulated speciation time behaves as $\tau_{sw}^* e^{f(CI/q^*)}$ with 'effective' $\tau_{sw}^* = \tau_{sw}/(1 + \log q/\log(s/m))$ and $q^* = q(1 + \log q/\log(s/m))$ picking up an additional logarithmic dependence on parameters, the exact origin of which is beyond our current approximations. This correction plausibly suggests rapid speciation, $\tau_{sw}^* \to 0$, when mutation rate become comparable to the selection strength $m/s \to 1$.

## Red Queen State is transient

We emphasize that the RQS regime in *Figure 3BC* is only transient. For any given $q$ and $d$, the RQS is likely to persist for a time given by the smaller of $\tau_{ext}$ and $\tau_{sp}$, before undergoing either extinction or speciation. These two processes limit the range of $q$ corresponding to the RQS from both sides in a time-dependent manner. *Figure 7* shows the likely state of an RQS system after time $\tau$ as a function of genetic diversity $q$ for the case of $d = 50$ and $\log(N_h \bar{I}) = 6.5$.

The regime of a single persistent lineage shrinks with increasing $\tau$, for example after $\tau = 10\tau_{sw}$ the RQS state likely prevails between $q = 1.5$ and $\approx 4$, while (for $d = 50$ and $\log(N_h \bar{I}) = 6.5$) it is unlikely to persist beyond $\tau \approx 100\tau_{sw}$ for any $q$. Both the maximal RQS lifetime and corresponding critical $q_c$, increase with increasing $d$.

## Discussion

The epidemiological and evolutionary dynamics of human RNA viruses show a number of qualitatively distinct patterns (*Grenfell et al., 2004*; *Koelle et al., 2011*). Agents of classical childhood diseases like measles or mumps virus show little antigenic evolution, other viruses like dengue- or norovirus exist in distinct serotypes, while seasonal influenza viruses undergo continuous antigenic evolution enabling viruses of the same lineage to reinfect the same individual.

Here, we have integrated classical multi-strain SIR models with stochastic models of adaptation to understand the interplay between the epidemiological dynamics and the accumulation of antigenic novelty. The former is dominated by the most prevalent strains, while the latter depends critically on rare pioneer strains that become dominant at later times. Our model differs from that of *Rouzine and Rozhnova (2018)* in two aspects that are crucial to questions addressed here: To meaningfully study speciation and diversification, the model needs to allow for an high dimensional antigenic space. Similarly, fluctuations in pathogen population size determine the dynamics of extinction and this aspect can not be studied in models with constant population size. Including these aspects of the epi-evolutionary dynamics allowed to define a 'phase' diagram that summarizes qualitatively different behavior as a function of the relevant parameter combinations, see *Figure 3B and C*.

The phase diagram shows which combinations of key parameters lead to three distinct outcomes: (1) extinction (red), (2) an evolving but low diversity pathogen population (yellow and green), (3) a deeply branching and continuously diversifying pathogen population (blue). The key parameters are the size of the population $\log(N_h s^2)$, the ratio of mutational effects to mutation rate $\log(s/m)$, and the cross-immunity range $d$. In particular, large $d$ prevents speciation, while rapid mutation and large population sizes facilitate speciation.

In regime (2) of a low diversity but rapidly evolving pathogen population, incidence is determined by the range of cross-immunity $d$ and by the speed of antigenic evolution which itself depends on the pathogen population size, mutation rates, and the fitness effect of novel mutations. A consistent solution of these dependencies shows that average incidence $I_{tot}$ decreases as $d^{-2}$, while weakly depending on population size and mutation rates (see *Equation A2.11*), consistent with results by *Rouzine and Rozhnova (2018)*. Typical values of the coalescent time of influenza A (2-4y), an infectious period of 5d, and a human population size $\sim 10^{10}$ result in an average annual incidence of 3–10%. This number is consistent with previous estimates of the annual attack rate of influenza (*Yang et al., 2015*) (which typically do not differentiate the different influenza lineages).

Of the different regimes, only extinction (1) and speciation (3) are truly asymptotic. The intermediate regimes of continuously evolving low diversity pathogen population - the Red Queen State (RQS) - are strictly speaking metastable states which eventually either go extinct or undergo branching, but in a certain regime of parameters are very long lived. In our simple model, stability against

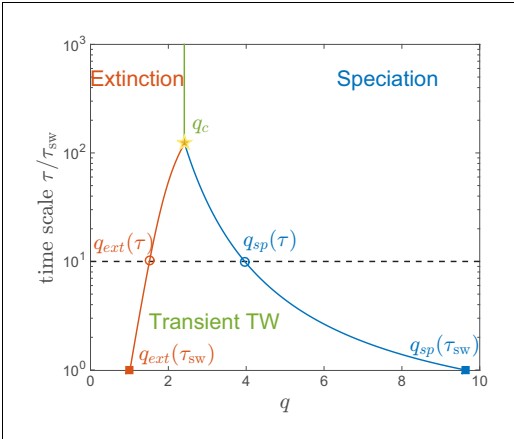

**Figure 7.** Lifetime of the RQS state. This schematic diagram based on *Figure 5A* defines the 'boundaries' of the transient RQS $q_{ext}(\tau)$ (red line) and $q_{sp}(\tau)$ (blue line). $q_{ext}(\tau)$ gives the value of $q$ for which the average time to extinction is equal to $\tau$ is defined similarly but for the speciation process. Average times to speciation and extinction become equal at the critical value $q_c$ at which RQS persists the longest. For $d = 50$, $q_c \approx 2.4$ and this value, along with the RQS lifetime, increases with $d$.
DOI: https://doi.org/10.7554/eLife.44205.012

speciation on the time scale $>10\tau_{sw}$ required $d \sim 10q$ (while stability against extinction requires $q > 2$). These results are consistent with earlier studies that have shown that competition between lineages mediated by long-range cross-immunity can prevent diversification, effectively canalizing the population into a single lineage (*Tria et al., 2005*; *Ferguson et al., 2003*).

In practice, the range of cross-immunity required to prevent speciation might be smaller than the idealized model. Our model assumes that the pathogen population can escape immunity via many equivalent mutational path. But in reality, the number of path to escape will be limited and some path more accessible than others, which will reduce the tendency to speciate and the necessity for large $d$. Similarly, other factors such as population turn over and geographic heterogeneity can delay extinction.

Previous studies have shown that the rate of branching in the speciation regime increases with population size and mutation rate consistent with the phase diagram (*Sasaki and Haraguchi, 2000*; *Koelle et al., 2011*). *Bedford et al. (2012)* have used large-scale individual-based simulations to explore structure of influenza viruses phylogenies. Consistent with our results, they found that the speciation rate increases with the mutation rate (lowering $\log s/m$ and thereby facilitating speciation) and increasing standard deviation of mutational effects. The latter increases the typical antigenic effect of successful mutations, which decreases the radius of cross-immunity when measured in units of mutations making the population more prone to speciate.

*Koelle and Rasmussen (2015)* have implicated deleterious mutation load as a cause of spindly phylogenies. Deleterious mutations increase fitness variation, which results in more rapid coalescence and less antigenic diversity, which in turn reduces speciation rates. Our model can readily incorporate deleterious effects of antigenic mutations on transmission $\beta$. Such deleterious mutations reduce the selection coefficient of antigenic mutations, which in turn reduces the fitness variance $\sigma^2$, see Appendix 6. After subtracting the contribution of deleterious mutations from the the fitness variance, the times to speciation follow the predicted dependence on $q$ and $d$, see *Figure 6C*.

Outbreaks of emerging viruses that quickly infect a large fraction of the population, as for example the recent Zika virus outbreak in the Americas, fall into regime (1): In 2–3 years, large fractions of the population were infected and have developed long-lasting immunity. As far as we know, the viral population did not evolve antigenically to escape this build up of herd immunity and the virus population is not expected to continue to circulate in the Americas (*O'Reilly et al., 2018*).

Different influenza virus lineages, in contrast, persist in the human population, suggesting that they correspond to parameters that fall into the RQS region of the phase diagram. Furthermore, the different subtypes display quantitatively different circulation and diversity patterns that allow for a direct, albeit limited, comparison to theoretical models: subtype A/H1N1 circulated with interruption from 1918 to 2009, A(H2N2) circulated for about 10 years until 1968, A/H3N2 emerged in 1968 and is still circulating today, and the triple reassortant 2009 H1N1 lineage, called A(H1N1pdm), settled into a seasonal pattern following the pandemic in 2009. Influenza B viruses have split into two separate lineages (B/Victoria and B/Yamagata) in the 1970s (*Rota et al., 1990*). Phylogenetic trees of A/H3N2 and the influenza B lineages are shown in *Figure 1*.

The influenza B lineages tend to be more genetically diverse than the influenza A lineages with a typical time to the most recent common ancestor of around 6 compared to 3 years, see *Figure 1*. Influenza A/H3N2 tends to have the lowest diversity and most rapid population turnover. This difference in diversity is consistent with influenza B lineages being more prone to speciation.

The typical diversity of these viruses needs to be compared to their rate of antigenic evolution. Hemagglutination inhibition titers drop by about 0.7–1 log2 per year in A/H3N2 compared to 0.1–0.4 log2 per year for influenza B lineages (*Smith et al., 2004*; *Bedford et al., 2014*; *Neher et al., 2016*). Hence the ratio of the time required to lose immunity and $T_{MRCA}$ is similar for the different lineages, suggesting that the distinct rates of genetic and antigenic evolution can not be used as a straight forward rationalization of the speciation event of Influenza B and the lack of speciation of influenza A lineages. Nor should such an explanation be expected as there is only a single observation of speciation. We note that currently circulating A/H3N2 viruses are exceptionally diverse with a common ancestor that existed about 8 years in the past. Furthermore, the cocirculating 3c.3a and 3c.2a are antigenically distinct and it is conceivable that further antigenic evolution will result in speciation of A/H3N2 viruses.

While we have shown that the natural tendency of SIR models to oscillate couples to the instability of the nose of the pathogen fitness distribution, making a quantitative link to the observed epidemiological dynamics of the flu is difficult on account of seasonal oscillation in transmissivity. The latter confounding factor is widely believed to be the cause behind observed seasonality of the flu. Including explicit temporal variation (in $\beta$) in our model would lock the frequency of the prevalence oscillation to the seasonal cycle, possibly resulting in subharmonic modulation, yet distinguishing such a modulation on top of an already stochastic process is hard. Much remains to be done: finite birth rates, distinct age distributions (as for example is the case for the two influenza B lineages), realistic distribution of antigenic effect sizes, or very long range T-cell-mediated immunity would all be interesting avenues for future work.

## Materials and methods

### Clone-based simulations

We simulate the original model on a genealogical tree by combining the deterministic update of SIR-type equations and the stochastic step introducing mutated strains. In each time step $\Delta t < 1$, we apply the mid-point method to advance SIR equations *Equations (1,2,4)*. We then generate a random number uniformly sampled between zero and one for each surviving strain with $N_h I_a \geq 1$. If the random number is smaller than $m N_h I_a \Delta t$ for strain $a$, we append a new strain $b$ as a descendent to $a$. The susceptibility to strain $b$ is related to susceptibility to strain $a$ via $S_b = (S_a)^{e^{-1/d}}$. In most of the simulations, the transmissibility of different strains is held constant $\beta$. Otherwise we allow for a strain specific transmissibility that is to its parent: $\beta_b = \beta_a - \delta\beta$ with $\delta\beta > 0$ for the deleterious effect of antigenic mutations and $\beta_b = \beta_{\max}$ if the mutation is compensatory. The new strain grows deterministically only if $\beta_b S_b > 1$.

This simplified model contains six relevant parameters: transmissibility $\beta$, recovery rate $\nu$, mutation rate of the virus $m$, birth/death rate of the hosts $\gamma$, the effective cross-immunity range $d$, and the effective size of the hosts $N_h$, whose empirical ranges are summarized in the *Table 1*. For flu and other asexual systems in RQS, $\beta \gtrsim \nu \gg m, \gamma, d \gg 1$, and $N_h \gg 1$.

Simulation code and output are available on github in repository FluSpeciation of the neherlab organization (*Neher and Yan, 2019*; copy archived at https://github.com/elifesciences-publications/FluSpeciation).

### Fitness-class-based simulations

The stability of the RQS and the extinction dynamics is fully captured by the traveling wave *Equation (8)*. We simulate the traveling wave by discretizing the fitness space $x$ into bins of step size $s$ around zero. The number of individuals infected by different strains correspond to integers in each bin $x_i$. At each time step, the population in each bin $I_i$ updates to a number sampled from the Poisson distribution with parameter $\lambda_i = N_h I_i (1 + (x_i - \bar{x})\Delta t)$ determined by mean fitness $x_i$ and a dynamic mean fitness $\bar{x}$, which increases by $\Delta t I_{tot}$, where $I_{tot}$ is the total infected fraction summed over all bins. When $\bar{x}$ becomes larger than one bin size $s$, we shift the all populations to left by one bin and reset $\bar{x}$ to , a trick to keep only a finite number of bins in the simulation. At the same time, antigenic mutation is represented by moving the mutated fraction in each bin to the adjacent bin on the right. The fraction is determined by a random number drawn from the Poisson distribution with the mean

**Table 1.** Relevant quantities of influenza virus and parameters in multi-strain SIR model.

| Symbol | Definition | Typical values for influenza | Range in simulations |
|---|---|---|---|
| $I_a$ | Fraction of population infected with strain $a$ | | |
| $S_a$ | Population average susceptibility to strain $a$ | ~0.5 | |
| $I_{tot} = \sum_a I_i$ | Total prevalence | | |
| $\bar{I}$ | Average total prevalence | 0.005 | |
| $d$ | Cross-immunity range | | [50, 200] |
| $\beta = \nu R_0$ | Transmission rate | ~0.5/day | 2 |
| $\nu$ | Recovery rate | ~0.2/day | 1 (sets unit of time) |
| $\gamma$ | Host birth/death rate | ~0.01/year | 0 |
| $K_{ab} = e^{-|b-a|/d}$ | Cross-immunity of strains $a$, $b$ | | |
| $\tau_{sw}$ | Coalescent time scale/sweep time | ~2-6 years | |
| $T_{MRCA}$ | Time to most recent common ancestor | ~2-10 years | |
| $\delta T$ | Fluctuations of $T_{MRCA}$ | ~2-6 years | |
| $s$ | Selection coefficient | ~0.03/week | [0.003, 0.05] |
| $m$ | Beneficial mutation rate per genome | $10^{-3}$/week | [$10^{-7}$,$10^{-3}$] |
| $N_h$ | Host population | $10^{10}$ | [$10^6$, $10^{12}$] |

DOI: https://doi.org/10.7554/eLife.44205.013

$mI_i\Delta t$. The typical ranges of the three parameters $s$, $m$, and $N_h$ follow the parameters in the genealogical simulation, as documented also in *Table 1*.

## Stochastic differential-delay simulation

To simulate the differential delay equations *Equations (11-13)*, we discretize time in increments of $\Delta t = \tau_{sw}/k$ and update the dynamical variables $\chi_i = x_n(t_i)$ and $\eta_i = I_{tot}(t_i)$ via the simple Euler scheme:

$$\chi_{i+1} = \chi_i + \Delta t(\chi_i - \eta_i) + \frac{\chi_i}{qs}\sqrt{\Delta t}\xi_i; \tag{14}$$

$$\eta_{i+1} = \bar{I}\exp\left(\tau_{sw}\chi_{i-k} - \frac{\tau_{sw}^2}{k}\sum_{j=0}^{k}j\eta_{i-j}\right), \tag{15}$$

where $\xi_i$ is a Gaussian random variable with zero mean and unit variance. Mean prevalence, $\bar{I}$, enters as the control parameter (which defines the time average of $\eta_i$).

## Influenza phylogenies

Influenza virus HA sequences for the subtypes A/H3N2, A/H1N1, A/H1N1pdm, as well as influenza B lineages Victoria and Yamagata were downloaded from fludb.org.

We aligned HA sequences using mafft (*Katoh et al., 2002*) and reconstructed phylogenies with IQ-Tree (*Nguyen et al., 2015*). Phylogenies were further processed and time-scaled with the augur (*Hadfield et al., 2018*) and TreeTime (*Sagulenko et al., 2018*). The analysis pipeline and scripts are available on github in repository 2019_Yan_flu_analysis of the neherlab organization.

## Additional information

### Competing interests

Richard A Neher: Reviewing editor, *eLife*. The other authors declare that no competing interests exist.

## Funding

| Funder | Grant reference number | Author |
|---|---|---|
| Simons Foundation | 326844 | Boris I Shraiman |

The funders had no role in study design, data collection and interpretation, or the decision to submit the work for publication.

## Author contributions

Le Yan, Richard A Neher, Conceptualization, Resources, Software, Formal analysis, Investigation, Methodology, Writing—original draft, Writing—review and editing; Boris I Shraiman, Conceptualization, Formal analysis, Funding acquisition, Investigation, Methodology, Writing—original draft, Writing—review and editing

## Author ORCIDs

Le Yan (iD) https://orcid.org/0000-0003-1323-0545
Richard A Neher (iD) https://orcid.org/0000-0003-2525-1407
Boris I Shraiman (iD) https://orcid.org/0000-0003-0886-8990

## Decision letter and Author response

Decision letter https://doi.org/10.7554/eLife.44205.024
Author response https://doi.org/10.7554/eLife.44205.025

## Additional files

### Supplementary files

• Transparent reporting form DOI: https://doi.org/10.7554/eLife.44205.014

### Data availability

Computer programs used for numerical simulations and analysis have been made publicly available at https://github.com/neherlab/FluSpeciation (copy archived at https://github.com/elifesciences-publications/FluSpeciation).

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

## Appendix 1

DOI: https://doi.org/10.7554/eLife.44205.015

### Approximation of susceptibility

A microscopic model that tracks the infection history of every individual in population is computationally costly and impossible to analyze analytically. To gain insight, we and other authors before us have used approximations that reduce the exploding combinatorial complexity of the state space (*Kryazhimskiy et al., 2007*). Here, we explore and justify the two separate approximations we have made to arrive at *Equation 2*: We ignore correlations between subsequent infections of the same individual and approximate the multiplicative effect of all subsequent infections by an exponential term.

To derive *Equation 4* we start with *Equation 3* and expand it in powers of $K$

$$
\begin{aligned}
S_a &= \langle \prod_b (1 - K_{ab}\sigma_b(i)) \rangle_i \\
&= 1 - \sum_b K_{ab}\langle\sigma_b(i)\rangle + \frac{1}{2}\sum_{c,b} K_{ab}K_{ac}\langle\sigma_b(i)\sigma_c(i)\rangle + ...
\end{aligned}
\tag{A1.1}
$$

where angular brackets denote the average over all individuals $i$ in the population and $\sigma_b(i) \in [0,1]$ denotes whether individuals $i$ was infected with strain $b$ in the past. This expansion assumes $|K_{ab}| \ll 1$ which would hold uniformly for weak inhibition $\alpha \ll 1$ but also holds for perfect inhibition for sufficiently distant strains $a, b$. For $\alpha \approx 1$ the greatest cause of concern is the contribution of the most proximal strain, to which we shall return later. To evaluate the terms on the right-hand-side we note that $\langle\sigma_b(i)\rangle = R_b$, that is the fraction of the population recovered from $b$, and $\langle\sigma_b(i)\sigma_c(i)\rangle = R_b R_c + \rho_{bc}$ where $\rho_{bc}$, by definition, is the correlation between infection with $b$ and $c$ at the level of individuals. Our approximation – following the well established logic of 'Mean Field' theories – neglects $\rho_{bc}$ compared to $R_b R_c$ (*Landau and Lifshitz, 2013*; *Weiss, 1907*). In this case, correct to order $|K|^2$, we can re-exponentiate the right-hand-side obtaining *Equation 4*. This simple derivation effectively captures the content of the 'order-1 independence closure' in *Kryazhimskiy et al. (2007)*.

Several facts about influenza in human populations suggest that the weak-correlation approximation is a reasonable starting point for modeling population scale behavior. (i) Seasonal flu epidemics involve a large number of strains, a particular strain infects only a small fraction of the population. Hence the $R_a$ are small and correlation effects are of minor importance. (ii) Challenge studies have shown that protection through vaccination or infection with antigenically similar strains is moderate and a large fraction of challenged individuals still shed virus (*Clements et al., 1991*). This possibility of homotypic re-infection shows that all $K_{ab}$ are substantially smaller than 1, supporting our approximation of population wide susceptibility, as discussed above. (iii) Antibody responses are polyclonal and differ between individuals such that the cross-immunity matrix is stochastic at the level of individuals. This variation in the cross-immunity matrix further reduces correlations in infection history at the population level and justifies the mean field approach taken here (*Lee et al., 2019*). (iv) Correlation in infection history induced by immunity are further reduced by the variation in exposure history through geography and variation in contact networks.

To quantify the error made by these approximations in the worst case scenario, we explore the case of a one-dimensional strain space with strictly periodic re-infection as soon as the virus population as evolved by $\epsilon d$ – the case of maximal correlation. The susceptibility of an individual last infected with a strain $x < \epsilon d$ mutations away from the current strain has susceptibility

$$
\begin{aligned}
S(x) &= \prod_{m=0}^{\infty}(1 - K(x + m\epsilon d)) \\
&= (1 - K(x))e^{\sum_{m=1}^{\infty} \log 1 - K(x + m\epsilon d)} \\
&\approx (1 - K(x))e^{-\alpha e^{-x/d} \sum_{m=1}^{\infty} e^{-m\epsilon}} \\
&= (1 - K(x))e^{-\alpha e^{-\frac{x}{d(e^{\epsilon}-1)}}}
\end{aligned}
\tag{A1.2}
$$

where we separated the most recent infection from previous infections to explicitly compare our approximations to that of **Rouzine and Rozhnova (2018)**. The only approximation so far happened between step 2 and 3. In the following, we will include the most recent infection in sum in the exponential to obtain the weak-inhibition approximation $\log S_{\mathrm{WI}} = -\alpha e^{-\frac{x}{d(1-e^{-\epsilon})}}$.

**Rouzine and Rozhnova (2018)** follow **Lin et al. (2003)** in approximating an individual's susceptibility by ignoring all but the most recent infection $S \approx 1 - K(x)$, thus keeping only the smallest term of the product representation of $S_a(i)$ in **Equation 3**. This approximation is referred to as 'minimum' cross-immunity in **Wikramaratna et al. (2015)**. Appendix 1—figure 1 compares the full expression and different approximations of $S(x)$ for different values of $\alpha$ and $\epsilon = 1$. For $\alpha = 1$, the 'most recent' approximation is better than the 'weak inhibition' approximation for $x < d/2$ but worse otherwise. For $\alpha < 1$, the weak inhibition approximation improves further.

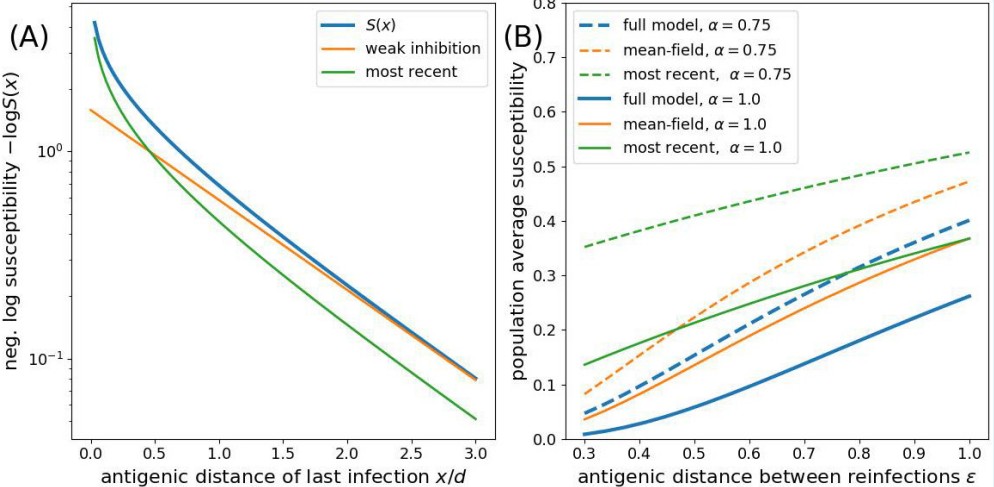

**Figure 1.** Approximations to the susceptibility. Panel A shows the effect of approximating the multiplicative effect of all past infection by $\log S \approx -\alpha e^{-\frac{x}{d(1-e^{-\epsilon})}}$ (weak inhibition) and by ignoring all by the most recent infection. Panel B shows the population average susceptibility assuming every individual gets reinfected once the virus has evolved by $d\epsilon$ for the multiplicative model, the mean field approximation, and the most recent infection approximation for different values of $\alpha$ as a function of the interval between infections $\epsilon$.

DOI: https://doi.org/10.7554/eLife.44205.016

To investigate the effect of ignoring correlations, we now compare the most correlated case of strictly periodic re-infection as soon as the pathogen has evolved by $\epsilon d$. For simplicity, we assume a time invariant density of recovered $1/d\epsilon$ (as in the analysis by **Rouzine and Rozhnova, 2018**). To calculate the population susceptibility, we integrate the expression for $S(x)$ for the full model and the 'most recent' approximation over the interval $[0, d\epsilon]$ and compare it to the mean field approximation in **Equation 4** with constant $R_b = 1/d\epsilon$. **Appendix 1—figure 1B** shows that the 'mean-field' approximation is closer to the full model across the entire range of relevant $\epsilon < 1$. Note that $\epsilon$ has to be determined self-consistently and will typically be of the order of the susceptibility.

Real-world influenza population are much less correlated then the extreme 'periodic infection' assumption used here for reasons listed above. The linearized mean-field approximation in *Equation 4* is therefore justified and can be expected to give a qualitatively correct approximation to a full model that tracks all infection histories.

## Appendix 2

DOI: https://doi.org/10.7554/eLife.44205.015

# Differential-delay approximation of RQS dynamics

Here we derive the differential delay system of equations that relate the behavior of the pioneer strains to the bulk of the population. Let us consider the generating function associated with the virus fitness distribution at time $t$:

$$G(\lambda, t) = \sum_i I_i(t) e^{\lambda x_i(t)} \tag{A2.1}$$

where $x_i(t) = x_n(t_i) - \int_{t_i}^t dt' I_{tot}(t')$ is the current fitness of the pioneer strain that first appeared at time $t_i$ and $I_i(t)$ is the fraction of the hosts infected by it:

$$I_i(t) = N_h^{-1} e^{\int_{t_i}^t dt' x_t(t')} = N_h^{-1} e^{x_n(t_i)(t-t_i) - \int_0^{t-t_i} dt' t' I_{tot}(t-t')} \tag{A2.2}$$

We next take a coarse grained view of pioneer strain establishment replacing the sum in **Equation (13)** by an integral over initial times $t_i \to t - \tau$

$$G(\lambda, t) = \frac{1}{N_h} \int_0^\infty \frac{d\tau}{\tau_a(t-\tau)} e^{(\tau+\lambda)x_n(t-\tau) - \int_0^\tau dt'(t'+\lambda)I_{tot}(t-t')}, \tag{A2.3}$$

where $1/\tau_a(t-\tau)$ is the rate at which new clones are seeded at time $t - \tau$. Let us evaluate the integral in the saddle approximation which is dominated by $\tau = \tau^*$ corresponding to the maximum in the exponential

$$\tau^* + \lambda = \frac{x_n(t-\tau^*)}{x_n'(t-\tau^*) + I_{tot}(t-\tau^*)} \approx \tau_{sw} \tag{A2.4}$$

where we have used the deterministic limit of **Equation 12**. To simplify presentation we shall ignore the time dependence of $\tau_{sw} = s^{-1} \log(x_n/m)$ replacing $x_n(t - \tau^*)$ in the logarithm by the time average $\bar{x}_n$.

Within the saddle approximation we then have

$$\log N_h G(\lambda, t) \approx x_n(t - \tau_{sw} + \lambda)\tau_{sw} - \int_0^{\tau_{sw}-\lambda} dt'(t'+\lambda)I_{tot}(t-t'), \tag{A2.5}$$

where we have omitted the logarithmic corrections for simplicity. Note that by definition $G(0, t) = I_{tot}(t)$.

We can now estimate fitness mean

$$\begin{aligned}
\bar{x}(t) &= \frac{d}{d\lambda} \log G(\lambda, t)\big|_{\lambda=0} \\
&= \tau_{sw}[x_n'(t-\tau_{sw}) + I_{tot}(t-\tau_{sw})] - \int_0^{\tau_{sw}} dt' I_{tot}(t-t') \\
&= x_n(t-\tau_{sw}) - \int_0^{\tau_{sw}} dt' I_t(t-t')
\end{aligned} \tag{A2.6}$$

and variance

$$\begin{aligned}
\sigma^2(t) &= \frac{d^2}{d\lambda^2} \log G(\lambda, t)\big|_{\lambda=0} \\
&= \tau_{sw}[x_n''(t-\tau_{sw}) + I_{tot}'(t-\tau_{sw})] + I_{tot}(t-\tau_{sw})
\end{aligned} \tag{A2.7}$$

**Equation (A2.7)** involves the second derivative $x_n''$ and we therefore expect fluctuations in the establishment of new lineages (which contribute to $x_n'$) to be quite important. Yet we can get useful insight by using the deterministic approximation to $x_n$ dynamics in **Equation 12**, in which case we arrive at simple delay relation between the variance and $x_n$

$$\sigma^2(t) = \tau_{\text{sw}}^{-1} x_n(t - \tau_{\text{sw}}) \tag{A2.8}$$

which is consistent with the variance calculated for the case of the steady TW and also satisfies the generalized Fisher theorem

$$\begin{aligned}
\frac{d}{dt}\bar{x} &= x'_n(t - \tau_{\text{sw}}) + I_t(t - \tau_{\text{sw}}) - I_t(t) \\
&= \sigma^2(t) - I_t(t)
\end{aligned} \tag{A2.9}$$

Combining **Equations 11, 12 and A2.8** we arrive at the deterministic dynamical system approximating coupled 'ecological' SIR dynamics with the evolutionary dynamics of antigenic innovation due to the pioneer strains.

$$\begin{aligned}
\frac{d^2}{dt^2}\log I(t) &= \tau_{\text{sw}}^{-1} x_n(t - \tau_{\text{sw}}) - I_{tot}(t) \\
\frac{d}{dt}x_n(t) &= \tau_{\text{sw}}^{-1} x_n(t) - I_{tot}(t)
\end{aligned} \tag{A2.10}$$

This system admits a family of fixed points of the form $\tau_{\text{sw}} I_{tot} = x_n = \bar{x}_n$, but as we show in C, the corresponding steady TW states are not always stable giving rise to limit cycle oscillations or leading to rapid extinction. The self-consistency condition relating $x_n$ and $I_{tot}$ for the steady traveling wave is readily generalized to limit cycle states. Integrating the differential-delay system over one cycle yields $\langle x_n \rangle = \tau_{\text{sw}}\langle I \rangle$. An additional relation is provided by integrating $\log N_h G(0, t)$ over the cycle:

$$\langle \log N_h I_{tot} \rangle = \frac{\tau_{\text{sw}}^2}{2}\langle I_{tot} \rangle \tag{A2.11}$$

A great deal of insight into the behavior of the (deterministic) differential delay system defined above is provided by its deterministic limit (see Appendix 3) which defines the stability 'phase diagram' shown in **Figure 3 (BC)** that correctly captures key aspects of the behavior observed in fully stochastic simulations.

# Appendix 3

DOI: https://doi.org/10.7554/eLife.44205.015

## Stability analysis of the differential-delay approximation

In the traveling wave case, it is natural to measure time in the units of the delay time scale $\tau_{\text{sw}}$. The therefore define a time variable $\zeta$ via $t = \tau_{\text{sw}}\zeta$, the fitness variable $\chi$ via $x_n = \tau_{\text{sw}}^{-1}\chi$ and the rescaled log-prevalence $u$ via $u = \log \tau_{\text{sw}}^2 I$ to obtain

$$
\begin{aligned}
\frac{d^2}{d\zeta^2}u(\zeta) &= \chi(\zeta-1) - e^{u(\zeta)} \\
\frac{d}{d\zeta}\chi(\zeta) &= \chi(\zeta) - e^{u(\zeta)}
\end{aligned}
\tag{A3.1}
$$

As before, this system has a one parameter family of fixed points $\chi = \bar{\chi}$, $u = \log \bar{\chi}$. Note that from the traveling wave model (**Desai and Fisher, 2007**), we have $\bar{\chi} = \bar{x}_n\tau_{\text{sw}} = q\log(x_n/m) = 2\log(N_h s^2)$. To analyze fixed point stability we linearize and Laplace transform, yielding

$$
\begin{aligned}
z^2\delta\hat{u}(z) &= e^{-z}\delta\hat{\chi}(z) - \bar{\chi}\delta\hat{u}(z) + z\delta u(0) + \delta u'(0) \\
z\delta\hat{\chi}(z) &= \delta\hat{\chi}(za) - \bar{\chi}\delta\hat{u}(z) + z\delta\chi(0)
\end{aligned}
\tag{A3.2}
$$

Stability is governed by the poles of the Laplace transformed response to the initial perturbation $\delta u(0)$, $\delta u'(0)$, $\delta\chi(0)$ and these poles are at the complex $z$ that solve:

$$
z = 1 + \bar{\chi}(1 - z - e^{-z})/z^2
\tag{A3.3}
$$

Fixed point - and hence steady RQS - stability requires $\Re(z) < 0$ which is found for $2 < \bar{\chi} < \bar{\chi}_c$. For $\bar{\chi} > 2.845$ one finds $\Im(z) \neq 0$ corresponding to the onset of oscillatory relaxation which turns into a limit cycle for $\bar{\chi} > \bar{\chi}_c \approx 16.6$. The period of the limit cycle is well approximated by $\Im(z)$, as the dashed line shown in the bottom panel of **Figure 4—figure supplement 1**.

The above stability analysis is done for the continuum limit $q \gg 1$. However the finiteness of $q$ does matter, especially close to extinction where only a small number of mutations separate most advanced strains from the bulk of the distribution. We shall now include the corrections to the first order in $1/q$. One such correction arises from the difference between the continuum $\chi(t_i)$ and discrete approximation of $\chi_i$, the position of the nose fitness bin relative to mean fitness at the time of its establishment. The other correction term comes via the establishment time $\tau_a$. Including both corrections the Langevin equation becomes:

$$
\frac{d}{d\zeta}\chi(\zeta) = \frac{1}{1 - \frac{1}{q}}\chi(\zeta) - \frac{1}{1 - \frac{1}{2q}}\frac{d^2}{d\zeta^2}u(\zeta) + \frac{1}{l}\xi(\zeta).
\tag{A3.4}
$$

To first order of $1/q$, the poles of the Laplace transform are determined by

$$
z^2 + \bar{\chi}\left[1 + \frac{1}{z-1}e^{-z} + \frac{1}{q}\frac{z^2}{(z-1)^2}e^{-z}\right] = 0.
\tag{A3.5}
$$

Solving for the onset of stability $\Re(z) = 0$, we find the extinction boundary $q_{ex}(\log N_h s^2)$ from the relation

$$
\bar{\chi} = \frac{2}{1 - \frac{2}{q_{ex}}} = 2\log(N_h s^2)
\tag{A3.6}
$$

We observe that $q_{ex} \to 2$ asymptotically for large $\log N_h s^2$.

## Appendix 4

DOI: https://doi.org/10.7554/eLife.44205.015

### Stochastic form of the differential-delay approximation

A sensible stochastic generalization is obtained by the stochastic approximation for the 'nose' dynamics in *Equation (12)*

$$\frac{d}{dt}x_n = \tau_{\mathrm{sw}}^{-1}x_n - I_t(t) + s\xi(t),$$  (A4.1)

combined with *Equation (A2.5)* at $\lambda = 0$

$$\log I(t) = \tau_{\mathrm{sw}}x_n(t - \tau_{\mathrm{sw}}) - \int_0^{\tau_{\mathrm{sw}}} dt' t' I_t(t - t').$$  (A4.2)

 Note that in this derivation we have avoided the need for explicitly approximating $\sigma^2$! (We have also neglected the effect of fluctuations arising from the logarithmic correction term effectively replacing it by its average value.) This stochastic differential delay (DD) system was used in simulations presented in *Figure 4C*.

## Appendix 5

DOI: https://doi.org/10.7554/eLife.44205.015

### Speciation rate as a stochastic 'First Passage' problem

Speciation occurs when two most distant clades persist until they are antigenically independent. This persistence problem can be formulated as a first passage problem by including the second 'nose' in the TW approximation.

We consider the birth of two pioneer strains at time $t = 0$, as illustrated in **Appendix 5—figure 1**. The descendants of the two strains forming two branches 1 and 2 diverge in the antigenic space as they persist in time. Suppose that at time $t$, the nose of branch 1 is at fitness $x_1$, and the nose of branch 2 is at $x_2$. Before the sweep time $t < \tau_{\mathrm{sw}}$, the cross-immunity grows mainly from the prevalent strains in the common ancestors of the two branches,

$$\frac{d}{dt}x_i = \tau_{\mathrm{sw}}^{-1}x_i - I_{\mathrm{tot}} + s\xi_i. \quad i = 1, 2 \tag{A5.1}$$

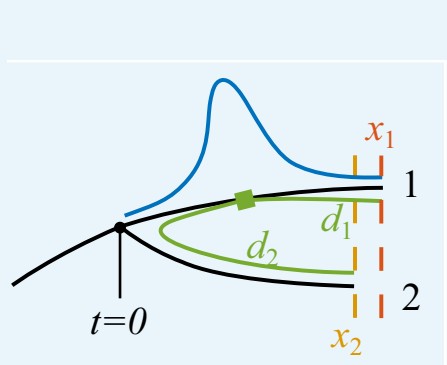
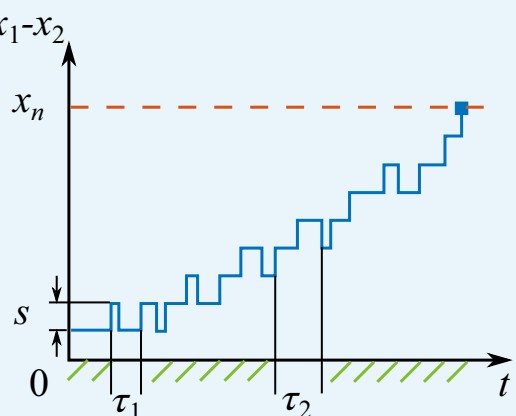

**Figure 1.** Process of speciation. Left: Sketch of a branching event at $t = 0$ with two branches 1 and 2. The fitnesses of the most fittest strains (noses) in branch 1 and 2 are $x_1$ and $x_2$. Branch 1 is the fitter one $x_1 > x_2$. The antigenic distances from the cross-immune bulk to the noses of the two branches are $d_1$ and $d_2$. The Gaussian profile in fitness is illustrated in blue. Right: The fitness difference between the two branches $x_1 - x_2$ is doing a biased random walk in time $t$ of step size $s$ with a reflecting boundary at $x = 0$ and an absorbing boundary at $x = x_n$.

DOI: https://doi.org/10.7554/eLife.44205.021

Later when $t > \tau_{\mathrm{sw}}$, the pathogen population splits and the different lineages evolve away from each other on two branches in the phylogeny. As the antigenic distances of each nose from the dominant strains on the own and the other branch differ, fitness of the two sets of pioneer strains changes at different rates:

$$\begin{aligned}
\frac{d}{dt}x_1 &= \tau_{\mathrm{sw}}^{-1}x_1 - I_1 e^{-d_{11}/d} - I_2 e^{-d_{21}/d} + S\xi_1; \\
\frac{d}{dt}x_2 &= \tau_{\mathrm{sw}}^{-1}x_2 - I_1 e^{-d_{12}/d} - I_2 e^{-d_{22}/d} + S\xi_2;
\end{aligned} \tag{A5.2}$$

where $d_{11}$ and $d_{22}$ scale roughly as $q$, the typical antigenic distance to the nose. In the limit $d_{21} \approx d_{12} \gtrsim d$, **Equation (A5.2)** reduce to two independent replicas of **Equation (12)** and the two branches are thus antigenically independent. What is the probability of reaching this limit? The approach to this question rather relies on the persistence probability of two branches in the other limit when $d_{21} \approx d_{12} \lesssim d$, where $I_1 + I_2 \approx I_{\mathrm{tot}}$ cross-immunity growth rate is approximately the same at both noses.

In this limit, the survival probability of the less fit nose maps to a first passage problem in the random walk of relative fitness $\zeta \equiv (x_1 - x_2)/x_n$. As illustrated in **Appendix 5—figure 1**, an establishment of nose one is a positive step of $\delta\zeta = s/x_n$, while an establishment of nose two results in a backward step of the same size. As the mutations arrive in characteristic times $\tau_1$ and $\tau_2$ depending on the nose fitnesses, in the continuum limit, we have

$$\frac{d}{dt}\zeta = \tau_{\text{sw}}^{-1}\zeta + \frac{s}{x_n}\xi,$$  (A5.3)

where $\xi$ is a random noise. There are two relevant boundaries: a reflecting boundary at $\zeta = 0$ where two branches switch roles in leading the fitness, and an absorbing boundary at $\zeta = 1$ where the fitness of less fit nose drops below the mean fitness and becomes destined for extinction.

The Langevin Equation in **Equation A5.3** corresponds to a diffusion equation for the probability density distribution $\rho(\zeta, t)$

$$\partial_t \rho(\zeta, t) = -\partial_\zeta[v(\zeta)\rho(\zeta, t)] + \partial_\zeta^2[D(\zeta)\rho(\zeta, t)],$$  (A5.4)

where the drift $v$ and diffusivity $D$ depend on $\zeta$,

$$v(\zeta) = \frac{1}{\tau_{\text{sw}}}\zeta; \quad D(\zeta) \approx \frac{1}{q\tau_{\text{sw}}}.$$  (A5.5)

Solving with boundary and initial conditions,

$$\begin{aligned}\partial_\zeta \rho(\zeta, t)|_{\zeta=0} &= 0; \ \rho(\zeta = 1, t) = 0; \\ \rho(\zeta, t = 0) &= \delta(\zeta),\end{aligned}$$  (A5.6)

we have

$$\rho(\zeta, t) = \sum_{n=1}^{\infty} e^{-\lambda_n t/\tau_{\text{sw}}} c_n \, {}_1F_1\left(\frac{1-\lambda_n}{2}, \frac{1}{2}, \frac{q}{2}\zeta^2\right),$$  (A5.7)

where ${}_1F_1$ is the generalized hypergeometric function, $\lambda_n$ is the $n$th smallest values solving ${}_1F_1(\frac{1-\lambda}{2}, \frac{1}{2}, \frac{q}{2}) = 0$, and coefficient $c_n$ is determined by the initial condition. In long time $t$, the slowest mode dominates the dynamics. In the large $q$ limit, we have $\lambda_1 = 1$. Since ${}_1F_1 \approx \text{const}$ for $\zeta \in (0, 1)$, the persistence probability is

$$P(T > t) \approx c e^{-t/\tau_{\text{sw}}}.$$  (A5.8)

The typical time interval between the establishment of successive pioneer strains at the nose scales as $\tau_a = \tau_{\text{sw}}/q$. We recall that speciation, or escape from cross-immunity, occurs when antigenic distance between the two branches in **Appendix 5—figure 1** $d_1 + d_2$ is larger than $d$. For that it suffices that the shorter branch $d_2 > d/2$ which occurs with probability

$$P(d_2 > d/2) \approx e^{-d\tau_a/2\tau_{\text{sw}}} = e^{-d/2q}.$$  (A5.9)

we find the probability of a successful branching $p_1$ to be proportional to $e^{-d/2q}$.

In the phylogenetic tree, $t/\tau_a$ trial branchings from the backbone arrive in time $t$. The probability that none of them successfully speciate is thus

$$P_{\text{nsp}}(t) = (1 - p_1)^{t/\tau_a} = e^{-t/\tau_{\text{sp}}},$$  (A5.10)

where the waiting time for speciation event is

$$\tau_{\text{sp}} \propto \frac{\tau_{\text{sw}}}{q} e^{d/2q},$$  (A5.11)

as numerically verified in **Figure 6**.

## Appendix 6

DOI: https://doi.org/10.7554/eLife.44205.015

### Effect of mutations on infectivity

Suppose an antigenic mutation has a deleterious effect on infectivity reducing the latter by $\delta\beta_d$ on average. This would effectively reduce the fitness gain of antigenic innovation from $s$ to $s_d(\beta) = s(\beta) - \Delta_d$, with $\Delta_d = \beta^{-1}\delta\beta_d$. In addition let us assume that there also are compensatory mutations which restore maximal infectivity $\beta_{\max}$. These compensatory mutation thus have a beneficial effect on fitness $\Delta_b(\beta) = \beta^{-1}\beta_{\max} - 1$. We assume that these mutations occur with rate $m_{\beta b}$. In a dynamic balance state the rate of fixation of compensatory mutations would exactly balance the deleterious mutation effect on $\beta$ so that $\tau_b^{-1}\Delta_d = \tau_a^{-1}\Delta_d$ with the fixation rate controlled by the fitness of the leading strain via $\tau_b^{-1} = x_n/\log(\frac{x_n}{m_{\beta b}})$. This dynamic balance is achieved at a certain value of $\beta_* < \beta_{\max}$, specifically $\beta_{\max} - \beta_* = \delta\beta_d\tau_b\tau_a^{-1}$ or $\beta_* = \beta_{\max} - \delta\beta_d r$ where $r = \log(\frac{x_n}{m})/\log(\frac{x_n}{m_{\beta b}})$.

The fitness of the nose of the distribution obeys

$$\frac{dx_n}{dt} = s_d(\beta)\tau_a^{-1} + \Delta_b\tau_b^{-1} - I_{tot} \tag{A6.1}$$

where the 1 st term on the RHS is rate of nose advancement due to antigenetic mutations $\tau_a^{-1} = x_n/\log(\frac{x_n}{m})$ as before, but with reduced fitness gain $s_d(\beta)$. The 2nd term describes the contribution of compensatory mutations. However in the dynamic equilibrium (at $\beta_*$) compensatory mutations exactly cancel the contribution the deleterious mutation contribution to $s$ so that for the steady state we recover

$$I_{tot} = s(\beta_*)\tau_{ag}^{-1} = \frac{s(\beta_*)x_n}{\log\frac{x_n}{m}} \tag{A6.2}$$

as we had for the TW driven by antigenic advancement only. The only effect is the reduction of $s$ from $s(\beta_{\max})$ to $s(\beta_*) = d^{-1}\log\beta_*$.

The sweep time, $\tau_{sw}$, upon which the fitness of the former pioneer strain comes down to the mean fitness and the nose fitness, $x_n$, retain the TW form

$$\tau_{sw} = \frac{x_n}{I_{tot}} = \frac{s(\beta_*)}{\log(\frac{x_n}{m})} \tag{A6.3}$$

Following TW approximation to estimate infection prevalence $\sqrt{I_{tot}} \sim N_h^{-1}\exp(x_n\tau_{sw}/2)$ as before one finds

$$x_n = 2\tau_{sw}^{-1}\log CN_h = 2s(\beta_*)\frac{\log[N_h s^2/\log(\frac{x_n}{m})]}{\log(\frac{x_n}{m})} \tag{A6.4}$$

The total fitness variance of the population contains a contribution, from antigenic mutations and the mutations in infectivity:

$$\sigma^2 = \tau_{sw}(s_d^2\tau_a^{-1} + \Delta_b^2\tau_b^{-1}) = x_n[s - 2\Delta_d + \frac{\Delta_d^2}{s} + \frac{\Delta_d\Delta_b}{s}] \tag{A6.5}$$

but under conditions of $\Delta_d$, $\Delta_b \ll s(\beta_*)$ total variance would also be decreasing.

Most relevant for our analysis however is not the typical, but the maximal antigenic distance within the viral population:

$$q_{ag} = \tau_{sw}\tau_{ag}^{-1} = \frac{x_n}{s(\beta_*)} = 2\frac{\log N_h s^2(\beta_*)c}{\log\frac{x_n}{m}} \tag{A6.6}$$

which is basically unchanged in the presence of infectivity mutations except for the expected reduction in the magnitude of $s^2$ factor inside the logarithm. Therefore, speciation rate would be reduced, but rather weakly, via a contribution subleading in $o(\log N_h)$

