## [Decision Letter]

Thank you for submitting your article "Phylodynamic theory of persistence, extinction and speciation of rapidly adapting pathogens" for consideration by *eLife*. Your article has been reviewed by three peer reviewers, one of whom served as a guest Reviewing Editor, and the evaluation has been overseen by Patricia Wittkopp as the Senior Editor. The reviewers have opted to remain anonymous.

The reviewers have discussed the reviews with one another and the Reviewing Editor has drafted this decision to help you prepare a revised submission.

Summary:

This manuscript analyzes a model of virus evolution in a host population in response to accumulating immune memory in previously infected individuals. The main result is a phase diagram that delineates qualitatively different modes of evolution (rapid extinction, strain proliferation, and metastable traveling fitness wave dynamics) as a function of evolutionary and epidemiological parameters.

Essential revisions:

The reviewers all agreed that the presented analyses were thorough and that the results were interesting. However, they also felt that several essential revisions were required:

1) The manuscript makes use of an existing status-based SIR model when mapping epidemiological dynamics on to the traveling wave evolutionary model. This model, first, should be explained in greater detail and with more clarity in the manuscript text. Further, this multi-strain model is one of two general types of multi-strain models (the other being a history-based model formulation). Are the phase diagram results robust to other SIR model formulations, including history-based formulations and the model formulation by Lin et al., 2003? Previous relevant work (Ballesteros et al. PLOS One) indicates that this might not be the case.

2) Text should be added to refer to several previous analyses that focus on very similar questions. Of particular importance is incorporating (to a much greater extent, both in the Introduction and starting around Equation 6) text that relates to the results presented recently in Rouzine and Rozhnova, 2018, ensuring that the overlap and the differences between these two analyses is accurately described. Koelle et al., 2011,) and Andreasen and Sasaki, 2006, also address similar questions about how certain epidemiological factors (population size, breadth of cross-immunity, etc.) affect whether and how quickly antigenic diversification will occur. The results presented here should be compared to those found earlier, even if these former approaches do not consider explicitly a traveling wave model. Finally, the way in which the epidemiological dynamics are mapped to fitness and how cross-immunity is quantified is identical to the approach outlined in Luksza et al., 2014, and this paper therefore needs to be cited, most notably in the context of Equations 1-3 and 4.

3) The mathematical conditions for the mapping onto fitness waves should be made more precise. This mapping is used throughout to describe the endemic regime. However, the traveling fitness wave formalism is derived under more specific assumptions, namely, the coexistence of many small-effect mutations (which corresponds to large values of q). Some parts of the phase diagram clearly fall outside this regime. While it may still be permissible to extend the asymptotic formulae, this should be discussed. We suggest to mark the boundary of the many-mutations regime, say, given by the condition U_b \gtrsim s, as a dotted line in the phase diagram.

4) In the Discussion, it would be important to emphasize that the metastability of the endemic regime is a result of the specific assumptions of this model and to discuss potential biological effects that alter the phase diagram. In particular, much work has been devoted to discuss mechanisms that stabilize the TW regime, e.g. the ideas of short-term broad cross-immunity (Ferguson et al., 2003, or random fitness components (Tria et al., 2005).

[Editors’ note: further revisions were requested before acceptance.]

Thank you for sending your article entitled "Phylodynamic theory of persistence, extinction and speciation of rapidly adapting pathogens" for peer review at *eLife*. Your article is being evaluated by two peer reviewers, and the evaluation is being overseen by a guest Reviewing Editor and Patricia Wittkopp as the Senior Editor.

The primary concern stems from details that are now provided in the revised manuscript that you submitted. Given these new details, reviewer #2 is particularly concerned that the epidemiological model does not incorporate infection histories appropriately, such that the results of this analysis do not advance the literature. The reviewer's request is that you re-do the entire analysis, using an epidemiological model structure that is appropriate. We would like to give you an opportunity to respond to this request, given the set of highly divergent approaches for modeling multi-strain dynamics.

Finally, the second reviewer requested that his/her entire initial review is transmitted in full. We will follow up shortly on the transmission of this entire review, as well as those from the two other reviewers.

We appreciate the value of simplified models; however, our previous comments we designed primarily towards making the simplifying assumptions explicit and to embed the epidemiological model better into the context of previous work on the subject, which we also discussed in the review consultation. Both issues are not yet adequately addressed in the current revision, and we regard the following points as essential for publication of this manuscript:

1) The assumptions underlying the epidemiological model, in particular with respect to the applicability to influenza, should be made explicit. It is not clear to us whether the model used here is indeed a generalization of previous models, as claimed. Specific points:

a) Some justification should be given for the steps leading from a general multi-strain immunity model to their Equations 1 – 3, e.g. along the lines of Rozhnova and Rouzine. For example, the authors could try to estimate, by the order of magnitude, the error of this approximation, at least, in a simple population configuration.

b) Equation 3 should be linked to the underlying dynamical model.

c) We also note again that the first application of this model to influenza data analysis (Lukza et al., 2014), which contains a very similar form of the equations and discusses their application to influenza, should be acknowledged in the context of Equations 1 – 3.

2) A quantitative comparison of the results of this paper to Bedford et al., 2012, and to Rozhnova and Rouzine should be given, for example in a supplementary figure. Specifically:

a) The fraction N_inf/N and average selection coefficient \σ, which allow the mapping to traveling wave theory, should be compared with previous work.

b) Also, it remains important to quantify the behaviour of the number of competing strains in the phase diagram (Figure 3B, C) in some fashion (see previous comment 3). The authors map the line q=1. Their reply otherwise refers to Figure 5, but it is not clear to us to which numbers q this refers to in Figure 3 (e.g., where is the locus q=10). Figure 3B, C shows two quantities as formulas in white font which we are not sure to which lines they refer to; please clarify and give units and numbers for these quantities.

Reviewer #2:

Unfortunately, not tracking the history of individual patients, i.e., not classifying patients according to previous infecting strains, is not a biologically correct approach, even though it has been done by two groups. This is not how the immune system works. One must track the memory cells from, at least, last infection. Virus infecting an individual reacts to memory in that individual, and not in other individuals. The oversimplification changes the results substantially and cannot be relied upon. For example, the turnover rate of population should not be an important parameter of the model. The dependence of the speed on parameters changes as well. We cannot be sure about the rest.

To avoid huge phase space, the simplest meaningful approximation is to track the last infection of an individual, i.e., to introduce the recovered uninfected individuals density and classify then according to memory cells left from their last infecting strain. One can show that older infection are a small correction. Then, consider multiple dimensions (analytically or numerically does not matter) and demonstrate that one-dimensional path arises automatically. After 1D path is assured, solve the 1D model analytically. Rouzine and Rozhnova did exactly that, in the case of the long-range immunity. Their multi-dimensional simulation is located in the end of Results and Supplementary Information. I also recommend to consult the previous numeric work of Bedford.

Therefore, I have to insist that the authors redo the work properly, with tracking the last memory of infection in individual. Otherwise, no numeric comparison is possible and cannot be in the future used for data comparison.

The original review follows for the authors' information:

The manuscript analyzes a model of the virus evolution in a host population due to accumulating immune memory in previously infected individuals. The authors use the SIR model by Gog et al., 2002, to map it to results of the traveling wave theory of evolution. If the cross-immunity between the virus and the memory is long-range, the authors demonstrate that the virus persists indefinitely. The state is a Red Queen process, a never-ending chase between virus and immune system in the antigenic space. If the cross-immunity is short-range, they find out that persistent infection is either unstable or splits into new states. An effective selection coefficient which makes the mapping to traveling wave possible is calculated.

The topic of the manuscript is important and the problem is challenging. The novel part of this work, compared to a recent paper on the same topic (below), is the comparison between long-range and short-range cross-immunity, and predicting the existence of a phase diagram of various behaviors including instability and oscillations.

I have some questions regarding the choice of the initial model, the sensitivity of results to its assumptions, and the connection to the previous work, as follows:

Major comments:

1) The SIR model is not explained in the manuscript, not the original paper by Gog et a. My questions are as follows.

a) According to my understanding, a typical infection is a stochastic event. An individual exposed to virus is either infected, at the systemic level, or not. If the individual is infected, the virus reaches high loads, causes a strong immune response, leaves high numbers of memory cells, and can be transmitted with appreciable probability to another individual. If the exposed individual is not infected at the systemic level, none of these events takes place. The probability of each of the two outcomes, given the exposure dose, depends on the presence of memory cells left from the previous infections, and their genetic distance from the infecting strain. Is this the scenario that the authors had in mind?

b) The model in MS considers a population structured into recovered and infected individuals classified by genetic variants of memory cells and virus, respectively. Are these population groups mutually excluding. What is their sum, the total population?

c) Can the authors draw a multi-compartment flow diagram of the model in supplement to show the processes they have included in the model?

d) An average adult person is infected by influenza virus more than once during lifetime. Indeed, between 4% and 20% individuals are infected annually. Therefore, all infections occur in previously infected (recovered) individuals. Yet, I do not see any infection of recovered individuals in model's equation. Who is infected then? This is especially confusing given that memory cells left from previous infections are the force that drives virus evolution.

f) What is the meaning of the exponential term in susceptibility S?

2) After Gog et al., 2002, Lin et al., 2003 have proposed an alternative, more transparent SIR model. Rouzine and Rozhnova, 2018 (RR) mapped that model to the traveling wave theory.

a) How does the change to Lin et al.'s version of SIR would affect the results on the stability of infection and the oscillatory states?

b) What is the difference with RR's results in the long range immunity case?

Additional comments:

3) In contrast to authors' statement, neither them nor RR's included the fluctuations of population size. If the authors implied that RR substituted the total population size instead of infected population to the traveling wave theory, they are mistaken: RR did the same rescaling.

4) The main difference between two models is in the choice of the initial SIR model (see above). RR considered the case of long-range cross-immunity only.

5) I would write the equations for the effective selection coefficient and for the rescaling of population size in separate lines, since they are important mapping formulas.

Reviewer #3:

The revised version has adequately addressed most of the comments, except the following:

I still think it would be relevant to quantify the behaviour of the number of competing strains in the phase diagram (Figure 3B, C) in some fashion (see previous comment 3). The authors map the line q=1. Their reply otherwise refers to Figure 5, but it is not clear to me to which numbers q this refers to in Figure 3 (e.g., where is the locus q=10). Figure 3B, C shows two quantities as formulas in white font which I am not sure to which lines they refer to; please clarify and give units and numbers for these quantities.

With this amendment, I think the paper is ready for publication.

---

## [Author Response]

Essential revisions:The reviewers all agreed that the presented analyses were thorough and that the results were interesting. However, they also felt that several essential revisions were required:1) The manuscript makes use of an existing status-based SIR model when mapping epidemiological dynamics on to the traveling wave evolutionary model. This model, first, should be explained in greater detail and with more clarity in the manuscript text. Further, this multi-strain model is one of two general types of multi-strain models (the other being a history-based model formulation). Are the phase diagram results robust to other SIR model formulations, including history-based formulations and the model formulation by Lin et al., 2003? Previous relevant work (Ballesteros et al. PLOS One) indicates that this might not be the case.

We have provided more details in the derivation of susceptibility model (Equation 3) adding a discussion which connects it to “status” and “history” based models. Our approximation is based on a factorization of the probability of different immune histories of individual hosts, which directly relates to the approach of Kryazhimskiy et al., 2007, in enabling a drastic reduction of the state space relative to the original “status” and “history” models. This approximation retains the dependence of the susceptibility of the host population to the prior history of infections, without tracking infection histories of individuals. Our resulting model of susceptibility is analogous to that derived in Kryazhimskiy et al., 2007, and the one used, prior to that, by Gog and Grenfell, 2002. These connections are fully acknowledged in references. We have also added references to and comments on Lin et al. and Ballesteros et al. Ballesteros et al. present four different two strain models (history vs status, reduced infectivity vs reduced susceptibility) and compare simulation results for these models. Only one (the status based reduced infectivity model) shows recurrent waves of infection by the mutant strain at high levels of cross-immunity. However, the parameters for cross-immunity of these models cannot be compared quantitatively. The differences between the simulations are thus due to choices of values of parameters that are phenomenological in nature and don’t have a one-to-one correspondence to reality. In the multi-strain case, we expect all models to exhibit qualitatively similar dynamics with appropriately chosen parameters – especially after reducing the high-dimensional history of status space to a linear number of variables, see Kryazhimsky et al.

2) Text should be added to refer to several previous analyses that focus on very similar questions. Of particular importance is incorporating (to a much greater extent, both in the Introduction and starting around Equation 6) text that relates to the results presented recently in Rouzine and Rozhnova, 2018, ensuring that the overlap and the differences between these two analyses is accurately described.

We now discuss the work by Rouzine and Rozhnova (R&R) at greater length. R&R map a multi-strain model in a one-dimensional antigenic landscape to a TW models of population genetics and the formulation of the model as well as the mapping to TW models is analogous to our approach. In contrast to R&R, our point of departure is a model in a high dimensional antigenic space and we use this model to show how the effectively one-dimensional TW emerges rather than introducing this as model assumption. R&R use their model to infer parameters through explicit comparison to influenza diversity data, which we don’t attempt. Instead, we use this model to explore the processes of extinction and speciation which can’t be studied in one-antigenic dimension with constant population size examined in R&R.

Koelle et al., 2011, and Andreasen and Sasaki, 2006, also address similar questions about how certain epidemiological factors (population size, breadth of cross-immunity, etc.) affect whether and how quickly antigenic diversification will occur. The results presented here should be compared to those found earlier, even if these former approaches do not consider explicitly a traveling wave model.

We have added the two suggested references along with a brief description in the Discussion section.

Finally, the way in which the epidemiological dynamics are mapped to fitness and how cross-immunity is quantified is identical to the approach outlined in Luksza et al., 2014, and this paper therefore needs to be cited, most notably in the context of Equations 1-3 and 4.

We now explicitly refer to Luksza et al. in this context.

3) The mathematical conditions for the mapping onto fitness waves should be made more precise. This mapping is used throughout to describe the endemic regime. However, the traveling fitness wave formalism is derived under more specific assumptions, namely, the coexistence of many small-effect mutations (which corresponds to large values of q). Some parts of the phase diagram clearly fall outside this regime. While it may still be permissible to extend the asymptotic formulae, this should be discussed. We suggest to mark the boundary of the many-mutations regime, say, given by the condition U_b \gtrsim s, as a dotted line in the phase diagram.

We have added a paragraph (bottom of column 1 on p 6) discussing typical values of *q* in influenza populations and making explicit comment that flu evolution is not likely to be in the asymptotic regime of large *q*. Nevertheless, we and others before us (references in the text), find that qualitative features of the TW models extend to modest values of *q* (as seen through comparison with direct simulation). Instead of adding a dotted line to indicate crossover in the phase diagrams of Figure 3B, C we added a note in the caption explaining that boundary of the”extinction” regime corresponds to *q* ∼*o*(1). More detailed view of the range of *q* corresponding to the RQS state is provided by Figure 5. We have added a remark about the range of *q* to the discussion of Figure 5 in the text, noting that (asymptotic) region of large *q* can only be reached in the limit of long-range cross-inhibition.

4) In the Discussion, it would be important to emphasize that the metastabilty of the endemic regime is a result of the specific assumptions of this model and to discuss potential biological effects that alter the phase diagram. In particular, much work has been devoted to discuss mechanisms that stabilize the TW regime, e.g. the ideas of short-term broad cross-immunity (Ferguson et al., 2003) or random fitness components (Tria et al., 2005).

In a strict sense, the endemic regime is always metastable as explicitly shown in Figure 5 which shows that RQS always goes away eventually either through extinction or through speciation. We have added text to further explain the meaning of Figure 5 to the reader. However the dependence of the extinction rate on model parameters might change through the addition of different model components. Instead of the logarithmic dependence of extinction rate on population size, this dependence might become polynomial or exponential in models that dampen population size fluctuations more strongly. We have added a short paragraph emphasizing that long range immunity and other features that dampen oscillations (population turn over, geographic structure, etc.) will tend to stabilize the RQS state.

[Editors' note: further revisions were requested prior to acceptance.]

Response to query regarding additional revisions:

Thank you for giving us an opportunity to respond to the points that came up after re-review of our manuscript. We are glad to read the reviewer #3 considers our revisions satisfactory and we can readily address the remaining request for clarification.

However, in response to our revised manuscript reviewer #2 now requests that we redo the analysis using a model that explicitly keeps track of the infection histories of all individuals. Based on previous work by several authors and our current understanding of influenza epidemiology and immunology, we have argued in the manuscript that this level of detail is not necessary and that the essential aspects of the dynamics are captured by simpler models that instead of individual infection histories keep track of susceptibilities to different strains in the population. We stand by this argument.

While infection history of an individual is important for predicting her/his susceptibility to infection by a given strain, the effective rate of spreading of that strain depends on the average susceptibility of individuals. This averaging makes our “mean field”-type approximation for population-wide susceptibility a natural first step. Let us restate here that there is a direct relation between the form of susceptibility that we use in the our work and the infection history description. It is easy to see that our expression for *S* given in Equation 3 is essentially exact in the limit of weak inhibition. The probability *p_a,i_*of individuals to become infected by strain *a* can be expressed as *p_a,i_*= ^Π^*_b_*(1 − *αK_ab_σ_bi_*) where binary *σ_bi_*∈ [0,1] denotes the infection history of individual *i* and *α* ≤ 1 sets the strength of inhibition (so that weak inhibition corresponds to *α* ≪ 1 is the cross-immunity kernel defined in the manuscript. Population wide susceptibility is the population average of *p_a,i_*of all individuals *i*:

⟨" close="⟩" separators="|">pa,i=⟨" close="⟩" separators="|">∏b1-αKabσbi=1-α∑bKab⟨" close="⟩" separators="|">σbi

+α22∑b∑c≠bKabKac⟨" close="⟩" separators="|">σbiσci

(1)

The term h*σ_bi_*i = *R_b_*is the fraction of people recovered from strain *b*. Correlations *ρ_bc_*between infections with strain *b* and *c* show up in the second order term h*σ_bi_σ_ci_*i = *R_b_R_c_*+ *ρ_bc_*. This simple derivation effectively captures the content of Kryazhimskiy et al., 2007, order-1 independence closure which assumes *ρ_bc_*= 0. We cite this work to make connection with prior work on the subject. In this case, we can simply exponentiate the expression to obtain our Equation 3, correct to order *α*^2^ (which is small for weak inhibition – we discuss the case of strong inhibition below):

Sa=⟨" close="⟩" separators="|">pai=e-∑bαKabRb (2)

We note that given this form of susceptibility and the homogeneity property of Equations 2-3 in the manuscript, parameter *α* can be eliminated by rescaling of *R* and *I* fractions, i.e. can be absorbed into effective host population size and does not explicitly appear in model analysis and simulation.

Several facts about influenza in human populations suggest that the weak inhibition approximation is a reasonable starting point for modeling population scale behavior.

- Seasonal flu epidemics involve a large number of strains, a particular strain infects only a small fraction of the population. Hence the *R_a_*are small and correlation effects are of minor importance.

- Challenge studies have shown that protection through vaccination or infection with antigenically similar strains is moderate and a large fraction of challenged individuals still shed virus [Clements et al., 1986]. This possibility of homotypic re-infection shows that *αK_ab_*are substantially smaller than 1, supporting our approximation of population wide susceptibility, as discussed above.

- Antibody responses are polyclonal and differ between individuals such that the cross-immunity matrix is stochastic at the level of individuals. This variation in the cross-immunity matrix further reduces correlations in infection history at the population level and justifies the mean field approach taken here.

- Correlation in infection history induced by immunity are further reduced by the variation in exposure history through geography and variation in contact networks.

Having provided the reasons why we think weak-inhibition approximation is appropriate, we note that the utility of (3) as a model of susceptibility does not end there! To wit, this approximation correctly captures the cross-inhibiting contribution of distant strains (on account of *K_ab_*for those strains being much less than 1, so that quadratic terms in *K* can be neglected compared to linear ones). Hence, even in the case of strong immunity *α* ≈ 1, only correlation terms involving close strains could contribute. In the traveling wave description of continuous adaptation, most relevant effects involve the spreading of the newly emerging antigenic variants in the “nose” of the fitness distribution. While these strains are antigenically close they occur at low frequencies, so that *R_a_* ≪ 1 and the correlations can again be neglected, the population-wide susceptibility to these strains is dominated by the effect of more distant strains (from further in the past).

Last but not least, expression (3) is an example of a “Mean Field Theory” type of approximations that replace the average of an exponential by an exponential of the average, neglecting correlation effects. This type of an approximation has an illustrious record of providing valuable insight into complex phenomena and are universally accepted in Physics: they are well recognized as the starting point for mathematical modeling.

In contrast, the proposal by reviewer #2 to simplify the problem by only tracking the last infection (that is dropping all terms in ^Π^*_b_*(1 − *K_ab_σ_bi_*) but the most recent one) is completely arbitrary and counterfactual. Fonville et al., 2014, have shown that immunity is maintained over decades and new infection results in a back-boost rather than a reset of the immunity landscape. It is also problematic as it would artificially facilitate speciation: It would increase susceptibility to viruses from sister clades since infection with one virus “wipes out” immune memory induced by a common ancestor.

Other points raised by reviewer #2 include:

**-** population turnover rate:the population turnover rate *γ* is NOT an important parameter of the model (as the reviewer pointed out) – and we never claimed it is. In fact, we explicitly set it to zero and it doesn’t feature in any of our conclusions.

- one- vs multidimensional trajectories:The reviewer suggests to “consider a multi-dimensional model […] and assure that a one dimensional path arises automatically. […] solve the 1D model analytically”. However, we show that a 1D path does not arise automatically and delineate conditions in which it does. Within these parameter ranges, we then investigate the model analytically as suggested by the reviewer. We are well aware of the work by Bedford et al., 2012, and Rouzine and Rozhnova, 2018. In Rouzine and Rozhnova, 2018, the 1D traveling wave is inherent in the model by either allowing for only one antigenic direction or imposing a preferred direction of antigenic escape (section 2.1 of the appendix of Rouzine and Rozhnova, 2018, – there seems to be inconsistent notation and a confusion of recovered and susceptible classes in the appendix).

- Fluctuations in population size:Our model explicitly accounts for fluctuations in the total number of infected individuals *I*_tot_. We don’t understand how the reviewer got the impression that our model assumes a constant (viral) population size.

Independent of the details of the epidemiological model, our work makes novel and important contributions to our understanding of pathogen evolution and dynamics:

- We show how qualitative features of epidemiological and evolutionary dynamics of rapidly adapting pathogens depend on parameters in a generic model of evolution in a high dimensional antigenic and genetic space. We delineate parameter regimes corresponding to speciation/diversification, single strain persistence, and extinction after a pandemic in a unified frame work. All previous analysis of this problem were either restricted to low dimensional spaces, a few strains, or purely numerical in nature.

- We connect the population genetics of rapid adaptation and with multistrain models of pathogens in a population that builds up immunity and show how epidemiological oscillation couple to the evolutionary dynamics of the pathogens. Rouzine and Rozhnova, 2018, don’t account for this crucial interaction – they consider a time invariant traveling wave solution of constant size.

The editor and reviewers have again carefully studied the previously revised manuscript and the authors' response to previous criticism. We appreciate the value of simplified models; however, our previous comments we designed primarily towards making the simplifying assumptions explicit and to embed the epidemiological model better into the context of previous work on the subject, which we also discussed in the review consultation. Both issues are not yet adequately addressed in the current revision, and we regard the following points as essential for publication of this manuscript:1) The assumptions underlying the epidemiological model, in particular with respect to the applicability to influenza, should be made explicit. It is not clear to us whether the model used here is indeed a generalization of previous models, as claimed.Specific points:a) Some justification should be given for the steps leading from a general multi-strain immunity model to their Equations 1 – 3, e.g. along the lines of Rozhnova and Rouzine. For example, the authors could try to estimate, by the order of magnitude, the error of this approximation, at least, in a simple population configuration.

We have added an Appendix 1 that explains in detail how a model based on individual infection histories reduces approximately to our “mean-field-theory” (MFT) type model of cross-immunity. We discuss the underlying assumptions in the light of known influenza immunology. Diversity of immune responses, non-perfect immune protection, and long-lasting immunity with known back-boost effects all suggest that the entire infection history is important at the level of individuals, but that population level dynamics is well described by a factorized distribution of histories. Following the suggestion of the decision letter, we evaluate the accuracy of our approximation (in Figure 8) for a specific “simple population configuration”, adopting for this purpose the scenario of periodic reinfection of individuals specifically considered by Rozhnova and Rouzine (R&R), which allows us to explicitly compare with the approximation used in their paper. In this R&R scenario, infection histories of individuals contain strong temporal correlations so that it may be expected to be a tough case for the MFT approximation (which neglects correlations in evaluating population averages). Nevertheless, we show (in Figure 8) that our approximation to the population averaged susceptibility is quite accurate in the relevant parameter regime. In particular, it is more accurate than the “most recent” approximation advocated for by one of the reviewers and used by Rouzine and Rozhnova.

b) Equation 3 should be linked to the underlying dynamical model.

Equation 3 is linked to the dynamical model by simple differentiation, which reproduces the corresponding equation in Gog and Grenfell, 2002 (in the limit of slow population turnover). This is now made explicit in Equation 4.

c) We also note again that the first application of this model to influenza data analysis (Lukza et al., 2014), which contains a very similar form of the equations and discusses their application to influenza, should be acknowledged in the context of Equations 1 – 3.

In response to the previous decision, we had included a reference to Lukzsa and Lassig, 2014, in the context of the cross-immunity function (prev Equation 4 as had been requested). The basic model defined in Equations 1-3 was introduced specifically in the context of influenza by Gog and Grenfell, 2002 and L&L refer to Gog and Grenfell for the definition of their model. We now explicitly point out that L&L used this model for influenza as well.

2) A quantitative comparison of the results of this paper to Bedford et al., 2012, and to Rozhnova and Rouzine should be given, for example in a supplementary figure. Specifically: a) The fraction N_inf/N and average selection coefficient \σ, which allow the mapping to traveling wave theory, should be compared with previous work.

We have added a discussion of the infected fraction predicted by our model. The values predicted by our model are compatible with observation and up to logarithmic factors agree with predictions by Rouzine and Rozhnova. Similar results hold for the average selection coefficient *s* (called *σ* in R&R), where we predict the same qualitative dependence on parameters. The main difference (we predict a weaker dependence on *R*_0_) can be traced to the “most-recent” approximation to the immunity history made by R&R. We show (see above) that our approximation is more accurate. We also discuss the results of Bedford et al., 2012m who studied dependence of genetic diversity and the tendency to speciate in large scale agent based simulations. Their observations are consistent with our analytic results.

We would like to stress, however, that the primary contribution of our work is *not*in an recapitulation of specific parameters of seasonal influenza virus, but in elucidating general properties of the coupling of evolutionary and epidemiological dynamics.

b) Also, it remains important to quantify the behaviour of the number of competing strains in the phase diagram (Figure 3B, C) in some fashion (see previous comment 3). The authors map the line q=1. Their reply otherwise refers to Figure 5, but it is not clear to us to which numbers q this refers to in Figure 3 (e.g., where is the locus q=10). Figure 3B, C shows two quantities as formulas in white font which we are not sure to which lines they refer to; please clarify and give units and numbers for these quantities.

We agree that our previous presentation on how *q* is related to the phase diagram was not optimal. We have now expanded our explanation of the regime boundaries in the phase diagram of Figure 3 specifically discussion the nature of the “critical” values of *q* (associated with these boundaries) and stressing how the transitions to extinction and speciation regimes depend on the time scale of observation. To this effect, we have added a subsection “Red Queen State is transient” (just before the Discussion section) and another figure (Figure 7) that explicitly shows the regimes of extinction and speciation in the plane of *q* and the observation time scale. We have also edited Figure 3B, C to clarify the labelling of the regime boundaries.